# QiMeng-NeuComBack: Self-Evolving Translation from IR to Assembly Code

**Hainan Fang**[1,2], **Yuanbo Wen**[1], **Jun Bi**[1], **Yihan Wang**[1,2], **Tonghui He**[1,2],
**Yanlin Tang**[1,2], **Di Huang**[1], **Jiaming Guo**[1], **Rui Zhang**[1], **Qi Guo**[1], **Yunji Chen**[1,2*]

[1]State Key Lab of Processors, Institute of Computing Technology, Chinese Academy of
Sciences, Beijing, China [2]University of Chinese Academy of Sciences, Beijing, China

{fanghainan23s, wenyuanbo, bijun, wangyihan25s, hetonghui23s,
tangyanlin21s, huangdi, guojiaming, zhangrui, guoqi, cyj}@ict.ac.cn

## Abstract

Compilers, while essential, are notoriously complex systems that demand prohibitively expensive human expertise to develop and maintain. The recent advancements in Large Language Models (LLMs) offer a compelling new paradigm: Neural Compilation, which could potentially simplify compiler development for new architectures and facilitate the discovery of innovative optimization techniques. However, several critical obstacles impede its practical adoption. Firstly, a significant lack of dedicated benchmarks and robust evaluation methodologies hinders objective assessment and tracking of progress in the field. Secondly, systematically enhancing the reliability and performance of LLM-generated assembly remains a critical challenge. Addressing these challenges, this paper introduces NeuComBack, a novel benchmark dataset specifically designed for IR-to-assembly compilation. Leveraging this dataset, we first define a foundational Neural Compilation workflow and conduct a comprehensive evaluation of the capabilities of recent frontier LLMs on Neural Compilation, establishing new performance baselines. We further propose a self-evolving prompt optimization method that enables LLMs to iteratively evolve their internal prompt strategies by extracting insights from prior self-debugging traces, thereby enhancing their neural compilation capabilities. Experiments demonstrate that our method significantly improves both the functional correctness and the performance of LLM-generated assembly code. Compared to baseline prompts, the functional correctness rates improved from 44% to 64% on x86_64 and from 36% to 58% on aarch64, respectively. More significantly, among the 16 correctly generated x86_64 programs using our method, 14 (87.5%) surpassed `clang-O3` performance. These consistent improvements across diverse architectures (x86_64 and aarch64) and program distributions (NeuComBack L1 and L2) validate our method's superiority over conventional approaches and its potential for broader adoption in low-level neural compilation.

## 1 Introduction

Compilers are indispensable and highly complex software systems, meticulously engineered over decades to translate high-level programming languages into low-level machine code executable by specific hardware architectures (Lattner & Adve, 2004). This process involves multiple stages of analysis, transformation, and optimization, often requiring deep expertise and significant development effort. In recent years, Large Language Models (LLMs) have made significant progress in natural language processing (OpenAI, 2024b; Liu et al., 2024; Yang et al., 2024a; Grattafiori et al., 2024)

---

[*]Corresponding Author.

39th Conference on Neural Information Processing Systems (NeurIPS 2025).

and increasingly excel at code-related tasks such as auto-completion, program synthesis, and bug detection, sparking interest in their broader software engineering applications (Hou et al., 2024; Wang et al., 2024; Rozière et al., 2024; Zhong & Wang, 2024).

Given their proficiency in handling complex code-related tasks, a natural and compelling research direction is the exploration of LLMs as core components in compilation (i.e., **Neural Compilation**). By directly translating high-level source code or intermediate representations (IR) into low-level assembly code using LLMs, *Neural Compilation* has the potential to augment or even replace traditional compiler stages. Compared to rule-based compilation, *Neural Compilation* has two appealing benefits. Firstly, it can drastically reduce the time and effort needed to build compilers for emerging Instruction Set Architectures (ISAs) (Armengol-Estapé & O'Boyle, 2021). Secondly, it can uncover new optimization strategies based on input program semantics due to LLM's ability to process code as lossless text (Cummins et al., 2023). Thus, *Neural Compilation* has the potential to transform how programming language researchers and hardware architects explore new designs.

Despite the escalating interest and conceptual feasibility demonstrated by early works (Armengol-Estapé & O'Boyle, 2021), *Neural Compilation* still faces major practical challenges, particularly in ensuring functional correctness and optimizing performance. First, achieving semantic equivalence comparable to traditional compilers remains a substantial challenge, even for LLMs trained on vast code corpora (Cummins et al., 2024). Second, while LLMs might uncover novel optimization strategies (Cummins et al., 2023), consistently surpassing the optimization levels of mature compilers (e.g., `clang-O3`) is an even greater hurdle. Moreover, the absence of systematic benchmarks for evaluating LLMs' *Neural Compilation* capabilities hinders objective progress assessment. Addressing this gap requires (1) a well-defined dedicated benchmark for consistent and measurable criteria, and (2) methods to improve the reliability and performance of LLM-aided neural compilation.

To this end, in this paper, we first introduce `NeuComBack`, a novel benchmark dataset specifically designed for evaluating IR-to-assembly compilation. Derived from `ExeBench` (Armengol-Estapé et al., 2022) and `TSVC` (Maleki et al., 2011), `NeuComBack` provides a diverse set of programs to systematically assess fundamental compilation and optimization capabilities. Note that the benchmark focuses on the critical backend compilation task of translating LLVM IR to hardware-specific assembly code across different ISAs. Using this benchmark, we conduct a systematic evaluation of state-of-the-art LLMs, including DeepSeek-R1 (Guo et al., 2025), establishing crucial performance baselines that were previously unavailable in this emerging research area. Second, we define a foundational *Neural Compilation* workflow, encompassing generation and optimization steps, which serves as a basis for our experiments and can inform future research in this domain. Building upon this workflow, we propose an automatic prompt learning method for assembly generation that learns from the LLM's self-debugging traces, systematically extracting insights from generation and correction attempts and iteratively evolving internal prompt strategies.

The main contributions of this work are summarized as follows:

- We introduce `NeuComBack`, a novel benchmark dataset tailored for the LLVM IR-to-assembly *Neural Compilation* task. Based on it, we conduct a comprehensive evaluation of state-of-the-art LLMs, establishing critical and previously unavailable baselines.

- We propose a self-evolving prompt optimization technique that enables LLMs to automatically improve their compilation capability through iterative refinement. By analyzing compilation errors and performance optimization trails, our method dynamically adapts prompt strategies, enhancing both the correctness and the performance.

- We conduct extensive experiments in terms of the functional correctness and the performance of LLM-generated assembly code. Compared to the baseline prompt, correctness increased from 44% to 64% (x86_64) and 36% to 58% (aarch64). More importantly, 14 of 16 correct x86_64 programs (87.5%) surpassed the performance of `clang-O3`. These results demonstrate consistent gains across diverse architectures and benchmarks, validating our approach's superiority in neural compilation.

# 2 Related Work

## 2.1 Neural Compilation

Machine learning has a long history in compiler optimization, traditionally focusing on heuristics for tasks like pass selection and phase ordering (Wang & O'Boyle, 2018; Agakov et al., 2006; Trofin et al., 2021). A more ambitious direction, *Neural Compilation*, aims for end-to-end assembly code generation. Early efforts in this vein employed neural machine translation techniques. For instance, Armengol-Estapé & O'Boyle (2021) pioneered direct C-to-x86 assembly translation using Transformers, and (Hosseini & Dolan-Gavitt, 2022; Armengol-Estapé et al., 2024) developed specialized models for decompilation.

The advent of LLMs has significantly advanced neural compilation. Cummins et al. (2023) demonstrated that LLMs, trained on millions of LLVM assembly and corresponding optimal compiler optimizations, could predict and perform the beneficial optimizations. Based on that, Cummins et al. (2024) further introduced LLM Compiler, a family of foundation models pretrained on compiler IRs and assembly. These models facilitate downstream tasks like compiler flag tuning and disassembly. LLMs are also increasingly applied to other compiler-level tasks such as sophisticated decompilation (Tan et al., 2024; Wong et al., 2023) and compiler fuzzing (Deng et al., 2023; Yang et al., 2024b). Fang & Mukhanov (2024) explored whether advanced reasoning mechanisms, like chain-of-thought (Wei et al., 2022), could enhance LLM performance in applying a single peephole optimization on AArch64 assembly. Their findings suggest that conventional LLMs, even when fine-tuned, struggle due to a lack of reasoning, and that models augmented with explicit reasoning capabilities (e.g., GPT-o1-preview) can significantly outperform them(Wang et al., 2023a).

Our work builds upon these diverse efforts in leveraging machine learning for compilation. While prior studies have demonstrated the initial feasibility of neural translation for compilation and showcased the growing power of LLMs in both optimizing and generating low-level code, achieving robust functional correctness and competitive performance simultaneously in *Neural Compilation* remains a primary research objective. As defined in Section 3, our research aims at translating LLVM IR to assembly code, targeting both functional correctness and performance optimization.

## 2.2 Automatic Prompt Learning

The performance of Large Language Models (LLMs) heavily depends on input prompts, making prompt engineering crucial. While manual techniques like zero-shot, few-shot prompting (Radford et al., 2019; Brown et al., 2020; Wei et al., 2024; Liu et al., 2022), and Chain-of-Thought (CoT) (Wei et al., 2022; Zhang et al., 2022; Zhao et al., 2024; Zhou et al., 2023; Wang et al., 2023b) exist, their laborious creation and often suboptimal or inconsistent results have spurred research into Automatic Prompt Optimization (APO).

APO methods algorithmically discover effective prompts through iterative refinement, sometimes using evolutionary approaches or LLMs as optimizers (Zhou et al., 2022; Pryzant et al., 2023; Yang et al., 2023; Guo12 et al.; Shum et al., 2024; Sun et al., 2023). Gao et al. (2025) introduced MAPS for test case generation, focusing on LLM-tailored prompts through diversity, domain knowledge, and failure-driven rule induction. Ye et al. (2025) proposed Prochemy for code generation/translation, using execution-driven iterative refinement to produce a fixed, reusable, optimized prompt. These highlight a trend towards automated, task-aware, and model-specific prompt engineering for code.

Our Automatic Prompt Learning (APL) method (Section 4) optimizes prompts for neural IR-to-assembly compilation, differing from general APO techniques by uniquely learning from the complete self-debugging trails. To the best of our knowledge, there are no research studies that specifically investigate the impact of automatic prompt learning on driving LLMs to generate assembly code. Experiments (Section 5) show that the proposed method effectively enables LLMs to learn from past practices, enhancing both the correctness and optimization quality of the generated assembly code.

# 3 Neural Compilation Task

## 3.1 Problem Formulation

Let $P_{source}$ denote a program expressed in a source representation, which can be either a high-level programming language $\mathcal{L}_{HL}$ (e.g., C, Python) or an Intermediate Representation (IR) $\mathcal{L}_{IR}$ (e.g., LLVM IR). Let $A_{target}$ be the corresponding program represented in a low-level assembly language $\mathcal{L}_{ASM}$ specific to a target hardware architecture $\mathcal{M}$ (e.g., x86_64, aarch64).

Formally, the task of **Neural Compilation** is defined as learning a mapping function $f_{NC}$, parameterized by a Large Language Model (LLM) with parameters $\theta$, which directly translates an input program $P_{source}$ into an output assembly program $A_{target}$: $A_{target} = f_{NC}(P_{source}, \mathcal{M}; \theta)$, $P_{source} \in \{\mathcal{L}_{HL}, \mathcal{L}_{IR}\}$, $A_{target} \in \mathcal{L}_{ASM}$.

This translation explicitly conditions on the target architecture $\mathcal{M}$, as assembly syntax and semantics are inherently architecture-dependent. A successful *Neural Compilation* system must satisfy two critical objectives simultaneously:

- **Functional Correctness**: The generated assembly program $A_{\text{target}}$ must maintain semantic equivalence to the source program $P_{\text{source}}$. Formally, this means that for any valid input or execution context, the observable behavior and final computational state must match precisely: $[\![A_{\text{target}}]\!]_{\mathcal{M}} \equiv [\![P_{\text{source}}]\!]$, where $[\![\cdot]\!]$ denotes the semantic interpretation of a program, and $[\![\cdot]\!]_{\mathcal{M}}$ specifically refers to execution semantics on the target hardware architecture $\mathcal{M}$.
- **Performance Optimization**: Beyond correctness, the generated assembly code should exhibit high performance. Performance metrics include execution speed (runtime), instruction count, code size, memory footprint, and energy consumption. Ideally, the assembly output from the LLM ($A_{\text{target}}$) should be competitive with or outperform code produced by highly optimized traditional compilers, such as those using the `-O3` flag: $c\big(A_{\text{target}}\big) \leq c\big(A_{\text{compiler(O3)}}\big)$, where $c(\cdot)$ denotes a performance cost function applicable to the target architecture.

Building on this formulation, we introduce a foundational workflow for *Neural Compilation* that systematically addresses both correctness and performance, drawing from common patterns in LLM-based code generation and optimization. The workflow follows the general structure below:

- **Initial Generation**: Given an input program $P_{\text{source}}$, the LLM generates an initial assembly candidate $A_{\text{target}}^{(0)}$ conditioned on the target architecture $\mathcal{M}$. If the initial generation fails to produce a functionally correct program, the process terminates early.
- **Iterative Optimization**: Starting from $A_{\text{target}}^{(0)}$, the model performs up to $T$ rounds of optimization. In each round $t = 1, 2, \ldots, T$, a new candidate $A_{\text{target}}^{(t)}$ is generated based on $A_{\text{target}}^{(t-1)}$, aiming to improve performance.

Note that after each generation or optimization step, a self-debugging procedure may be applied to verify and enforce functional correctness.

## 3.2 NeuComBack Dataset

### 3.2.1 Data Collecting

Our primary objective is to construct a comprehensive dataset for the LLVM IR to Assembly (ASM) compilation task. To the best of our knowledge, no publicly available datasets specifically focus on this IR-to-ASM translation. Consequently, we adapted existing C-to-ASM benchmarks. We selected `ExeBench` (Armengol-Estapé et al., 2022), a widely used collection of C programs, and the Test Suite for Vectorizing Compilers (TSVC) (Maleki et al., 2011), a compiler performance benchmark. The distinct characteristics of these sources naturally lead to a two-tiered dataset:

- **Level 1 (Fundamental Compilation, 200 tasks):** This selection from the `ExeBench` test suite covers a broad variety of real-world C programs and features diverse control-flow patterns. A significant portion of these programs, derived from embedded systems applications, exhibits intensive I/O operations. While extremely complex control flow (e.g.,

deep recursion, concurrency) is underrepresented in available sources, L1 is designed with a breadth-first focus to assess general-purpose functional correctness.

- **Level 2 (Optimization Potential, 151 tasks):** This level, drawn from the `TSVC` benchmark suite, features programs characterized by computationally simple execution paths but notable loop intricacy. Such programs provide a strong basis for assessing the optimization capabilities reflected in the generated assembly.

We curated two benchmark levels: NeuComBack-L1, comprising 200 `ExeBench` test cases selected for the longest IR lengths from a cleaned set of 1,618; and NeuComBack-L2, the entire `TSVC` benchmark (151 cases). While both levels have similar lines of code, Clang AST analysis shows `TSVC` programs average significantly more variables (26.06 vs. 13.73 for `ExeBench`). Crucially for LLM-based compilation, `TSVC`'s nested loop structures (prevalent in L2) present far greater difficulty than ifs or gotos due to complex register allocation and scheduling, making L2 a more demanding benchmark.

### 3.2.2  Data Cleaning

The original `ExeBench` dataset presented challenges in terms of code quality and complexity. Many programs were notably short (as illustrated in Appendix A, Figure 2), and the original authors employed `g++ -fpermissive` for compilation due to numerous deviations from C/C++ standards. To address this, we focused on the `ExeBench` test set, performing substantial rewriting to ensure C/C++ standard compliance. A critical step in our data cleaning pipeline was to rigorously verify that `clang` could successfully compile each modified program into LLVM IR.

Furthermore, to mitigate the issue of programs being overly simplistic or short, we filtered the cleaned `ExeBench` data. We specifically selected programs that, when compiled by `clang`, produced the longest LLVM IR sequences. This selection process enriches our Level 1 dataset with more substantial and representative examples for the IR-to-ASM task, resulting in a set of 200 programs.

### 3.2.3  Evaluation Metrics

The dataset adopts two primary metrics, designed to reflect the dual goals of *Neural Compilation*: achieving functional correctness and generating high-performance assembly code.

- **Functional Correctness (ACC):** This metric measures the percentage of test cases where the LLM-generated assembly code produces the exact same output as the reference assembly code compiled by `clang -O0` (or another appropriate baseline for correctness) for a given set of inputs. For the `TSVC` dataset, we enhanced its original correctness verification methodology. While the existing checks might suffice for traditional compilers, they are less adequate for LLMs, where errors can manifest at any instruction. Our improved validation is more rigorous in detecting such fine-grained discrepancies.

- **Correctness with Superior Performance (ACC+Perf):** This metric quantifies the percentage of programs that are not only functionally correct but also exhibit execution performance superior to that achieved by `clang -O3`. This directly assesses the LLM's capability to generate optimized assembly.

For `ExeBench`-derived cases, we reuse the dataset's official inputs and checkers, with only lightweight wrappers adapted to accommodate our IR→ASM setting. For `TSVC`, to avoid false positives from degenerate constant initializations, we replace fixed-value array initializations with deterministic pseudorandom values drawn under a fixed seed, providing diverse inputs while preserving reproducibility.

Performance evaluations are conducted exclusively on the Level 2 (`TSVC`) dataset, as the Level 1 (`ExeBench`) dataset, though containing diverse real-world programs, was not primarily designed for performance evaluation, making its programs unsuitable for such analysis. To ensure reliable performance measurements, we execute each program (both LLM-generated and the `clang -O3` baseline) 11 times. We then discard the first three (warm-up) and last three (cool-down) runs, retain runs 4–8 (middle five by execution order), and report the median of these five execution times (median computed by their execution time cost) to mitigate system noise and provide a stable performance figure.

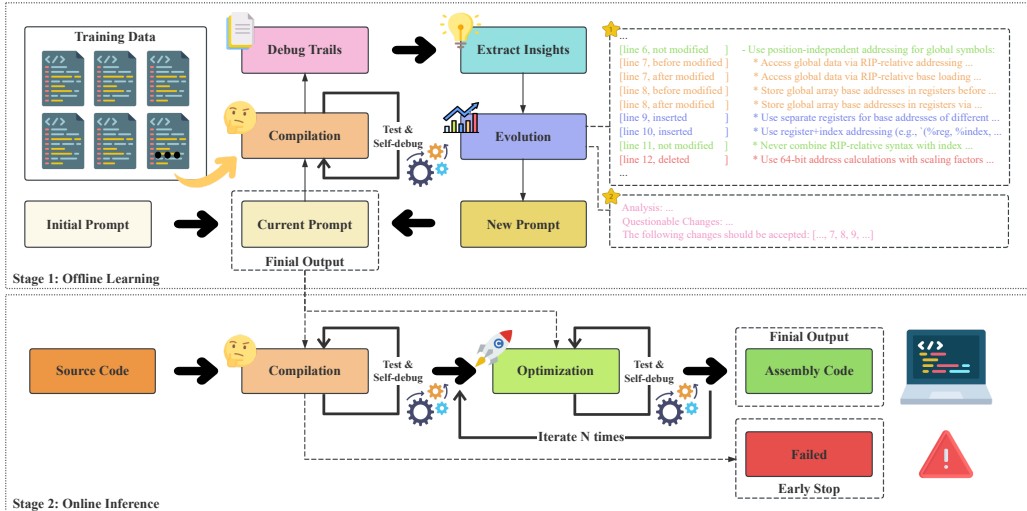

Figure 1: Pipeline of our automatic prompt learning method on *Neural Compilation*.

## 4 Method

### 4.1 Overview

Our *Neural Compilation* framework leverages a novel automatic prompt-learning mechanism specifically tailored to improve LLM-generated assembly code. Distinct from previous prompt evolution methods, our approach's core insight lies in learning from the LLM's complete iterative self-debugging process, which effectively enables the LLM to learn from its past practices in diagnosing and resolving complex errors in assembly code generation. Our method comprises two distinct stages, illustrated in Figure 1: offline prompt learning and online inference. In the offline stage, the model iteratively evolves prompts based on comprehensive analysis and insights derived from past generation trials. Subsequently, in the online stage, the model utilizes these refined prompts to generate and iteratively optimize assembly code, progressively improving quality and performance.

### 4.2 Self-Evolving Prompt Optimization for Neural Compilation

#### 4.2.1 Offline: Prompt Learning

The offline learning phase focuses on automatically discovering and evolving prompts to effectively guide the LLM toward generating correct and performant assembly code. This process begins with **initializing** an empty prompt template. Following this, **self-debug trials** are collected by using the LLM to perform a "compilation" process and then test the generated assembly translations; errors trigger iterative self-debugging, refining code until correctness or iteration limits are reached. Subsequently, **critical insights** are extracted by analyzing the complete self-debug trajectories to identify error patterns and effective strategies via LLM-assisted analysis. Finally, **prompt evolution** occurs by refining prompts through the integration of experience and insights (extracted from a batch of self-debug trials), which are then reviewed for clarity and effectiveness.

Concretely, we maintain a single evolving prompt $\pi$. During offline learning, after each mini-batch of compilation tasks we perform one update of $\pi$: we collect self-debugging trajectories from those programs that were initially incorrect but became correct via self-debug, and we ask the same model used for compilation to act as a "prompt optimizer." A meta-prompt instructs it to analyze the trajectories, extract error patterns and effective fixes, and propose edits to the existing prompt $\pi$ to reflect these new insights. After the LLM suggests revisions, a comparison and confirmation step follows: we prompt the LLM to review the change list and confirm only the necessary and beneficial edits, adding stability to the evolution process. The updated $\pi$ is then used for the next mini-batch.

#### 4.2.2 Online: Inference

The online inference stage deploys evolved prompts from offline learning to iteratively generate and optimize assembly code. First, the **initial assembly generation** is performed, using evolved prompts to generate initial assembly code from IR. This initial assembly is then tested, and iterative self-

debugging is triggered until functional correctness is achieved; if self-debug fails after all attempts, the generation is terminated, and a failure is reported. Next, **iterative optimization** is applied to further refine the initial assembly code, with evolved prompts still provided to minimize error introduction. Each optimization iteration includes testing and self-debugging as necessary to ensure continued correctness. Finally, the process **outputs** the optimized assembly code, demonstrating competitive or superior performance compared to compiler-generated code.

# 5 Experiments

## 5.1 Empirical Evaluation of Existing Large Language Models

To establish a performance baseline for current LLMs on the *Neural Compilation* task, we conducted an empirical study evaluating several state-of-the-art LLMs. The primary objective was to assess their capability in translating programs from an IR to correct and performant x86_64 assembly code. We utilized NeuComBack-L2 as the main test data for our experiment. For each of the 151 test cases, each LLM was tasked to generate x86_64 assembly code, involving an initial attempt followed by a maximum of two

Table 1: Baseline performance of advanced LLMs, NeuComBack-L2 (overall 151 cases), x86_64

| Model | ACC (%) | ACC+Perf (%) |
|---|---|---|
| GPT-4o | 1.99 (3/151) | 0.66 (1/151) |
| O3-Mini | 21.19 (32/151) | 5.30 (8/151) |
| O1 | 19.87 (30/151) | 5.30 (8/151) |
| DeepSeek-V3 | 14.57 (22/151) | 3.31 (5/151) |
| DeepSeek-R1 | 45.70 (69/151) | 21.85 (33/151) |

rounds of self-debugging. We evaluated five models: GPT-4o (OpenAI, 2024b), O3-Mini (OpenAI, 2025), O1 (OpenAI, 2024a), DeepSeek-V3 (Liu et al., 2024), and DeepSeek-R1 (Guo et al., 2025). The prompt for baseline assessment is detailed in Appendix B. The results are summarized in Table 1. DeepSeek-R1 demonstrated the strongest baseline, achieving 45.70% functional correctness and outperforming O3 in 21.85% of cases. Other reasoning models like O3-Mini and O1 also showed notable capabilities. Among models without specialized, extensive reasoning training, DeepSeek-V3 performed best.

The experimental results lead to several key observations. Firstly, advanced LLMs, particularly those enhanced with reasoning capabilities like DeepSeek-R1, exhibit substantially improved performance in the *Neural Compilation* task of IR-to-assembly translation compared to previous models (e.g., GPT-4o). DeepSeek-R1's ability to correctly generate assembly for nearly half of the benchmark cases, and for a significant portion of those to be more performant than -O3, is a promising development. Secondly, even for the best-performing model, achieving functional correctness remains a significant hurdle, and surpassing traditional compiler optimizations is an even greater challenge. The performance of DeepSeek-V3 suggests that foundational model capabilities are improving. These baseline results highlight both the potential of state-of-the-art LLMs in compiler tasks and the substantial room for improvement necessary for practical, widespread adoption.

## 5.2 Evaluation of Self-Evolving Prompt Optimization

Building on this empirical study, we evaluate the impact of our proposed automatic prompt learning method, using the DeepSeek-R1 model due to its superior performance. We assess its effectiveness on both NeuComBack-L1 and NeuComBack-L2, targeting the x86_64 architecture. The NeuComBack-L1 dataset was divided into 120 training sam-

Table 2: Performance of automatic prompt learning vs. baseline on NeuComBack-L1 (test set, 40 samples), x86_64, DeepSeek-R1.

| Method | ACC (%) |
|---|---|
| Baseline | 50.00 (20/40) |
| Our | 80.00 (32/40) |

ples, 40 for validation, and 40 for testing, while NeuComBack-L2 comprised 101 training samples, 25 for validation, and 25 for testing. For both datasets, we conducted prompt learning over three epochs with a batch size of 5, additionally introducing 1 self-debugging round per generation for NeuComBack-L1 and 2 rounds for NeuComBack-L2. From this process, we selected the highest-performing prompt based on validation metrics (referred to as the "Learned Prompt"), which we compare against the baseline prompt detailed in Appendix B.

Consequently, for NeuComBack-L1, we evaluate functional correctness (ACC) as its programs, derived from `ExeBench`, serve as fundamental test cases for basic compilation correctness. For NeuComBack-L2, sourced from `TSVC` and featuring programs with higher loop complexity and optimization potential, we assess both ACC and ACC+Perf (correctness with superior performance over `clang -O3`).

**Results on NeuComBack-L1.** As shown in Table 2, the application of the learned prompt led to substantial improvements: the number of functionally correct solutions increased from 20 (50.00% ACC) with the baseline prompt to 32 (80.00% ACC), with a relative 60% improvement.

**Results on NeuComBack-L2.** We conducted two sets of experimental setups: (a) using a baseline prompt for both initial generation and subsequent optimization, and (b) using the learned prompt (see Appendix C), performing up to two self-debug rounds after each generation or optimization step." Table 3 details the results on the NeuComBack-L2 benchmark ("–" indicates no change compared to the previous stage). In the initial generation phase, our method increased correctly solved programs on the NeuComBack-L2 test set from 11 (44.00% ACC) with the baseline prompt to 16 (64.00% ACC), achieving a 45% relative improvement, and solutions outperforming -O3 increased from 6 to 10 (a relative improvement of approximately 67%) When evaluating the full *Neural Compilation* workflow, we observe that the advantages of the learned prompt are not merely preserved but substantially amplified through iterative optimization. The baseline prompt produced 7 solutions (28% ACC+Perf) that outperformed the O3 optimization level, whereas the learned prompt consistently generated 14 such solutions (56% ACC+Perf), yielding a 100% relative improvement in high-performance assembly code generation. This significant improvement arises from the learned prompt's ability to guide the model effectively across the entire compilation process. By steering the model away from common suboptimal decisions and error patterns, the learned prompt enhances the optimizer's overall effectiveness, leading to consistently superior results.

Table 3: Performance of automatic prompt learning vs. baseline prompt on NeuComBack-L2 (test set, 25 samples), x86_64, DeepSeek-R1.

| Method & Stage | ACC (%) | ACC+Perf (%) |
|---|---|---|
| **Baseline Prompt** | | |
| *After Initial Generation* | 44.00 (11/25) | 24.00 (6/25) |
| *After 2 Rounds Iter. Opt.* | – | 28.00 (7/25) |
| **Our Method (Learned Prompt)** | | |
| *After Initial Generation* | 64.00 (16/25) | 40.00 (10/25) |
| *After 2 Rounds Iter. Opt.* | – | 56.00 (14/25) |

A noteworthy fact is that, among the 16 programs correctly generated using learned prompts, 14 (87.5%) were further optimized to surpass `clang-O3` performance (Detailed speedups are reported in Appendix D.). This remarkably high conversion rate demonstrates that when properly guided through iterative refinement, LLMs can achieve not just functional correctness but genuine optimization mastery in complex assembly-level tasks, a capability that appears substantially underutilized in current approaches. Furthermore, the consistent performance improvements across both datasets validate the robustness and general applicability of our prompt learning methodology, suggesting its potential for broader adoption in low-level code optimization scenarios.

Meanwhile, the consistent positive impact across both datasets also underscores the robustness and general applicability of our learning approach.

## 5.3 Ablation

**Method Effectiveness on Different Instruction Sets.** To evaluate adaptability to different hardware, we experimented with the aarch64 instruction set using the DeepSeek-R1 model and our NeuComBack-L2 dataset (101 train, 25 validation, 25 test). The learning process involved 3 epochs with 4 self-debugging rounds per generation. The prompt was chosen the same. As detailed in Table 4, this approach significantly enhanced performance on the test set: functional correctness doubled from 9 to 18 solutions (a 100% relative improvement), and solutions surpassing -O3 performance increased from 2 to 7 (a 250% relative improvement), demonstrating effectiveness for aarch64.

Table 4: Effectiveness of automatic prompt learning, DeepSeek-R1, NeuComBack-L2 (test set, 25 samples), aarch64

| Method | ACC (%) | ACC+Perf (%) |
|---|---|---|
| Baseline | 36.00 (9/25) | 8.00 (2/25) |
| Our | 72.00 (18/25) | 28.00 (7/25) |

**Transferability of Learned Prompts across Different Data Distributions.** We investigated prompt transferability by applying an x86_64 prompt learned on NeuComBack-L2 directly to NeuComBack-L1 (120 training, 40 validation, 40 test samples) without further learning. As shown in Table 5, the NeuComBack-L2-transferred prompt achieved 67.50% functional correctness on the NeuComBack-

Table 5: Performance on NeuComBack-L1 (x86_64, DeepSeek-R1) using different prompt strategies, showing test set and overall ACC

| Prompt Strategy | Test Set ACC (%) | Overall ACC (%) |
|---|---|---|
| Default Prompt | 50.00 (20/40) | 54.50 (109/200) |
| Learned on NeuComBack-L1 | 80.00 (32/40) | - |
| Learned on NeuComBack-L2 | 67.50 (27/40) | 74.50 (149/200) |

L1 test set, a 35% relative improvement over the default prompt's 50.00%. While still lower than the 80.00% from prompt learned specifically on NeuComBack-L1, it demonstrates positive transferability, suggesting the learned prompts can capture generalizable patterns across program distributions.

**Effectiveness in Reducing Self-Debug Rounds.** We analyzed the average number of self-debug rounds for tested programs that were eventually solved correctly. This evaluation was performed across different architectures and datasets, comparing our method against the baseline. As shown in Table 6, our method consistently reduces the average number of self-debug rounds needed. These results indicate that, in addition to achieving higher final functional correctness (as demonstrated in previous sections), our automatic prompt learning approach enables the LLM to converge on correct assembly code more efficiently, requiring fewer self-correction attempts.

Table 6: Average self-debug rounds for successfully compiled programs by DeepSeek-R1 on test sets, comparing our method with the baseline. Lower is better.

| Architecture | Dataset | Max Debug Rounds | Avg. Self-Debug Rounds | |
| --- | --- | --- | --- | --- |
| | | | Baseline | Our Method |
| x86_64 | NeuComBack-L1 | 1 | 0.90 | 0.28 |
| | NeuComBack-L2 | 2 | 1.09 | 0.25 |
| aarch64 | NeuComBack-L2 | 4 | 1.44 | 1.22 |

## 6  Case Studies

**Analysis of LLM-generated Code on Performance**  As shown in our experiment (Section 5), LLMs demonstrate a promising ability to optimize assembly code, sometimes achieving performance even superior to traditional compilers. Below, we present examples that illustrate the specific optimization techniques employed by LLM. For instance, LLMs can enhance performance by reducing the instruction count. Consider the NeuComBack-L2 function `s452` as an example, shown in Figure 3. While both the original LLVM-generated version and the LLM-optimized version exhibit effective vectorization, the former contained redundant addition operations in its detailed implementation. These redundancies could be eliminated by pre-calculating constants. The LLM successfully implemented this optimization, notably reducing two `paddd` (packed doubleword add) instructions to a single one within the inner loop. This demonstrates the LLM's capability to iteratively refine and further optimize its own generated code.

Another illustrative example is `s332`, shown in Figure 4. In this case, the LLM achieves optimizations superior to those produced by LLVM's `O3` optimization level. When tasked with finding the first index of an array element satisfying an inequality condition, LLVM `O3` employs sequential comparisons within the inner loop, without leveraging vectorization. In contrast, the LLM leverages vector instructions such as `cmpps` (compare packed single-precision floating-point values). This enables the simultaneous comparison of four floating-point numbers within the inner loop. Furthermore, it employs the `bsfl` (bit scan forward, least significant) instruction to efficiently extract the index from the resulting comparison mask, thereby achieving functional equivalence with the original C code but with significantly improved performance.

**Analysis of the Learned Prompts**  The effectiveness of our automatic prompt learning method hinges on the quality and specificity of the evolved prompts. To shed light on what the LLM learns and incorporates into its guiding instructions, this subsection provides a content analysis of the learned prompts. A complete example of a learned prompt, illustrating these aspects in full, can be found in Appendix C.

In our iterative prompt engineering process, the LLM was progressively enhanced by integrating a comprehensive set of summarized rules. These rules were meticulously designed to address various dimensions of code generation, aiming to systematically elevate the quality, consistency, and accuracy of the generated outputs. By codifying best practices into structured guidelines, the process not only refined the LLM's syntactic and semantic understanding but also established a robust framework for maintaining alignment with specific coding standards and conventions.

These rules spanned multiple aspects of code generation, as shown in Table 7: Firstly, formatting rules were established, mandating critical principles

Table 7: Cases of rules learned

| | |
| --- | --- |
| Formatting | Placing function code exclusively in .text section |
| | Positioning .size directives immediately after function bodies |
| Syntatic | Using .L prefix with exact spelling for local labels |
| | Append @PLT suffix to external function calls (e.g., 'call dummy@PLT') |
| Semantic | Clear return registers (XORL %eax,%eax) for void functions |
| | Pass stack-based parameters in reverse order with alignment padding |

like the distinct separation of data and code segments, and the precise IEEE 754 binary representation for floating-point numbers. Secondly, syntactic rules were introduced to govern structural correctness, exemplified by the specifications for XMM register indexing and conventions. Thirdly, semantic rules provided guidance and best-practice recommendations regarding the nuanced usage of particular instructions, thereby improving the LLM's understanding of instruction semantics.

# 7    Conclusion

This paper advances the field of Neural Compilation by introducing a dedicated IR-to-Assembly benchmark and a novel self-evolving prompt optimization method, enabling more systematic and rigorous evaluation of LLM-based compilation techniques. We first introduce NeuComBack, a benchmark specifically designed for IR-to-assembly compilation, and conduct a comprehensive evaluation of state-of-the-art LLMs to establish performance baselines. We further propose a self-evolving prompt optimization method that enables LLMs to iteratively improve their compilation strategies by learning from self-debugging traces. Our experimental results demonstrate significant improvements in both functional correctness (increasing from 44% to 64% on x86_64 and from 36% to 58% on aarch64) and optimization performance (with 87.5% of correct x86_64 programs surpassing `clang-O3`). These consistent gains across multiple architectures and benchmark distributions validate the effectiveness of our approach and its potential to advance low-level neural compilation.

While our method achieves promising results, there remains room for improvement in both functional correctness and optimization performance. Future work will focus on further refining these LLM-driven compilation techniques and expanding benchmarks to encompass more complex real-world compilation scenarios.

# 8    Acknowledgement

This work is partially supported by the Strategic Priority Research Program of the Chinese Academy of Sciences (Grants No.XDB0660300, XDB0660301, XDB0660302), NSF of China (Grants No.62525203, 62302483, U22A2028, 6240073476), the Science and Technology Major Special Program of Jiangsu (Grant No. BG2024028), CAS Project for Young Scientists in Basic Research (YSBR-029) and Youth Innovation Promotion Association CAS.

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

## A    Original ExeBench C Code (Test Set) Statistics

The original `ExeBench` dataset, prior to our cleaning and filtering process, contained a large number of C programs. As discussed in Section 3.2 (specifically the Data Cleaning subsubsection), many of these programs were notably short. Figure 2 provides a visual representation of the C code line count distribution within the original `ExeBench` dataset (test set).

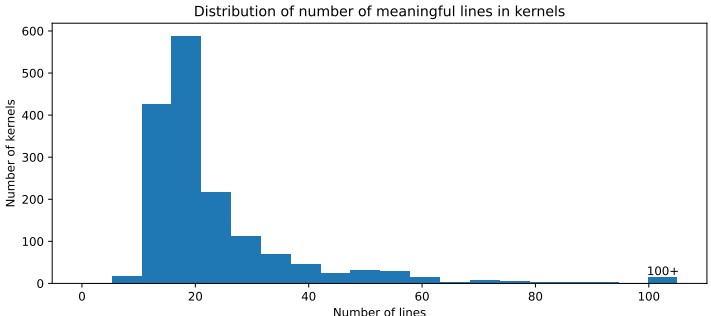

Figure 2: Original `ExeBench` C code statistics. This figure illustrates the distribution of C code lines in the programs from the original `ExeBench` dataset (Test Set).

## B    Baseline Prompt

This appendix shows the baseline prompt utilized for the x86_64 architecture. This prompt serves as the initial, unoptimized set of instructions given to the Large Language Model for translating LLVM IR to AT&T format assembly code.

```
‘‘‘llvm ir
...
‘‘‘

‘‘‘assembly
...
‘‘‘

You are an expert in IR code and assembly code. Please help me translate the given
    IR code into x86_64 GNU assembly code (using AT&T format).

Note that:
1. You MUST use the following template to give out the complete, final target
    assembly code, and MUST NOT apply this template to any other part of your
    response:
‘‘‘assembly
...(Provide the assembly code here)
‘‘‘
2. Do not return any information other than the assembly code.
```

## C    Example of Learned Prompt

This appendix presents an example of a prompt that was evolved using our automatic prompt learning method. Specifically, this prompt is the result of the learning process conducted on the x86_64 architecture using the NeuComBack-L2 dataset, following the experimental practices described in Section 5.2.

```
‘‘‘llvm ir
...
‘‘‘

‘‘‘assembly
...
```

```
‘‘‘

You are an expert in IR code and assembly code. Please help me translate the given
    IR code into x86_64 GNU assembly code (using AT&T format).

Note that:
1. You MUST use the following template to give out the complete, final target
    assembly code, and MUST NOT apply this template to any other part of your
    response:
‘‘‘assembly
...(Provide the assembly code here)
‘‘‘
2. Do not return any information other than the assembly code.
3. Additionally, to guarantee the correctness of the generated assembly code, please
     ensure that:
   - Correctly structure sections and assembler directives by:
      * Placing function code exclusively in .text section and constant data in
    .rodata (or other appropriate sections)
      * Maintaining strict section boundaries no intermixing of code and data
    declarations between sections
      * Positioning .size directives immediately after function bodies within
    .text section, before any data/comm declarations or section transitions
      * Placing .comm directives exclusively in BSS context (typically .bss
    section), never within .text or .rodata
      * Using .L prefix with exact spelling for local labels and ensuring case-
    sensitive consistency in all references
   - Use position-independent addressing for global symbols:
      * Access all global data exclusively via RIP-relative addressing using ‘leaq
     symbol(%rip), %reg‘
      * Store global array base addresses in registers via LEAQ before indexed
    accesses, maintaining separate registers for distinct arrays
      * Use register+index addressing (e.g., ‘(%base_reg, %index, scale)‘) with
    scaling factors matching element size (4 for 32-bit floats/ints)
      * Never combine absolute symbol names with index registers; use base
    registers initialized via LEAQ for all indexed accesses
   - Maintain proper register usage and stack management:
      * Preserve callee-saved registers (rbx/r12-r15) via push/pop sequences when
    modified, with dedicated registers for persistent data like array bases and
    loop counters
      * Use 64-bit registers (rsi/rdi/etc) for loop counters and indices when
    handling ranges exceeding 32-bit capacity, initializing with XORQ for zeroing
      * Store loop counters in callee-saved registers separate from array pointers
    , using distinct registers for different control variables
      * Maintain 16-byte stack alignment by calculating adjustments as (
    pushed_registers*8 + parameters_size + 15) & ~15 before calls. Ensure alignment
     persists after push/sub operations, especially when storing XMM registers with
     movaps
      * Transfer final results to return registers (XMM0 for floats, RAX/EAX for
    integers) BEFORE restoring callee-saved registers in function epilogue
      * Clear return registers (XORL %eax,%eax) for void functions before RET and
    validate all exit paths set EAX/RAX
      * Explicitly extend 32-bit values to 64-bit via MOVSLQ/CDQE when using 32-
    bit operations with 64-bit registers
   - Handle function calls and external references correctly:
      * Append @PLT suffix to external function calls (e.g., ‘call dummy@PLT‘)
      * Preserve XMM registers XMM6-XMM15 across calls via save/restore sequences
    if reused post-call. When storing computed values in these registers, save
    originals to stack with movaps (aligned) or movups (unaligned), then restore
    before returning
      * Load parameters into registers via ‘leaq symbol(%rip)‘ immediately before
    call setup to minimize register pressure
      * Pass stack-based parameters in reverse order with alignment padding,
    recalculating offsets after stack adjustments to ensure correct argument
    positioning
```

```
        * Never assume XMM0-5 retain values across function calls; explicitly
    preserve results in callee-saved registers if needed
    - Maintain strict data type consistency:
        * Match IR operations with correct instruction types (e.g., use addss for
    float addition vs. addl for integers)
        * Represent immediate float constants as hexadecimal literals (e.g., 0
    x3f800000 for 1.0f) instead of separate memory constants when possible
        * Handle residual elements using same data type as vectorized operations (
    XMM registers for floats)
        * Use MOVSS for 32-bit float transfers between memory/XMM registers instead
    of integer MOV variants
    - Implement loop semantics accurately:
        * Place loop initializations inside headers when IR indicates per-iteration
    requirements (PHI-node dependencies)
        * Reinitialize array bases/values in outer loops if inner loops modify
    global state
        * Maintain separate vectorized (16-element) and scalar residual processing
    paths
        * Track register/memory modifications through nested loops to prevent cross-
    iteration dependencies
    - Handle vector operations correctly:
        * Use SIMD registers matching operation width (XMM for 128-bit ops)
        * Verify memory alignment against IR attributes before using aligned
    accesses (movaps)
        * Default to unaligned accesses (movups) when alignment isn't␣explicitly␣
    guaranteed
␣␣␣␣␣␣␣␣*␣Process␣residual␣elements␣after␣vectorized␣blocks␣using␣scalar␣operations␣
    matching␣original␣data␣type
```

## D   Detailed Runtime Performance Results

Table 8 reports per-program speedups of our method (learned-prompt, DeepSeek-R1) relative to `clang -O3` for the 16 functionally-correct programs referenced in Section 5.2 (after initial generation and 2 rounds of iterative optimization). Speedup is defined as $\text{Speedup} = \frac{\text{Runtime}_{O3}}{\text{Runtime}_{LLM}}$, so values $> 1$ indicate our code runs faster than `-O3`. Runtimes follow the same protocol as in Section 5: each program is executed 11 times and we take the median of the middle 5 runs.

Table 8: Per-program speedups (LLM over `clang -O3`) on NeuComBack-L2 (x86_64). Average speedup across these 16 programs is $1.28\times$; maximum is $3.25\times$ (s331).

| Program | Speedup |
|---------|---------|
| s331    | 3.2485  |
| s1232   | 2.0621  |
| s332    | 1.9954  |
| s112    | 1.1048  |
| s152    | 1.0852  |
| s2251   | 1.0543  |
| s451    | 1.0288  |
| s319    | 1.0278  |
| vdotr   | 1.0277  |
| s3112   | 1.0275  |
| s4121   | 1.0274  |
| s231    | 1.0253  |
| s113    | 1.0247  |
| s422    | 1.0224  |
| s452    | 0.8621  |
| s341    | 0.7924  |

# E Illustrative Examples of LLM-Driven Code Optimization

This appendix provides the visual representations (Figures 3 and 4) for the code optimization case studies discussed in Section 6. Figure 3 showcases the optimization of function `s452`, where the Large Language Model (LLM) reduced the instruction count. Figure 4 illustrates the optimization of function `s332`, where the LLM achieved performance superior to LLVM O3 by employing vector instructions. These figures compare the original LLVM-generated assembly code with the LLM-optimized version.

```
movdqa .LC0(%rip), %xmm8
movdqa .LC1(%rip), %xmm9
movdqa .LC5(%rip), %xmm10
...
paddd %xmm8, %xmm4
paddd %xmm9, %xmm4                    movdqa .LC0(%rip), %xmm10
...                                  movdqa .LC1(%rip), %xmm11
paddd %xmm8, %xmm5                    ...
paddd %xmm10, %xmm5                   paddd %xmm10, %xmm4
...                                  ...
.section .rodata                     paddd %xmm11, %xmm5
.align 16                            ...
.LC0:                                .section .rodata
.long 0, 1, 2, 3                     .align 16
.LC1:                                .LC0:
.long 1, 1, 1, 1                     .long 1, 2, 3, 4
.LC4:                                .LC1:
.long 4, 4, 4, 4                     .long 5, 6, 7, 8
```
(a) original version                              (b) optimized version

Figure 3: A case of performance self-optimization (`s452`)

```
movss      -16(%rbx,%rax,4), %xmm1
ucomiss        %xmm0, %xmm1
ja         .LBB0_10
movss      -12(%rbx,%rax,4), %xmm1
ucomiss        %xmm0, %xmm1
ja         .LBB0_9                     movups (%r14,%r13,4), %xmm0
movss       -8(%rbx,%rax,4), %xmm1
ucomiss        %xmm0, %xmm1            cmpps $6, %xmm1, %xmm0
ja         .LBB0_11                    movmskps %xmm0, %eax
movss       -4(%rbx,%rax,4), %xmm1     testl %eax, %eax
ucomiss        %xmm0, %xmm1            jnz .Lfound_in_vector
ja         .LBB0_12
```
(a) original version                              (b) optimized version

Figure 4: A case of performance better than O3 (`s332`)

# F Cost-Controlled Model Comparison

We first estimated the relative inference cost and found that a single experimental run with DeepSeek-R1 is approximately 14 times expensive than with DeepSeek-V3. Based on this finding, we conducted a cost-controlled evaluation. We allocated DeepSeek-V3 a 14-fold computational budget, allowing it to perform more sampling, to match the cost of a single run of DeepSeek-R1. All other experimental settings remain the same as those used in Section 5.2 on NeuComBack-L2. In our original experiments, DeepSeekR1 solved 11 out of 25 problems (achieving functional correctness). However, under this new cost-controlled setting, DeepSeekV3 successfully solved 17 out of 25 problems.

Table 9: Cost-controlled comparison on NeuComBack-L2 (test set, 25 cases).

| Setting | Problems Solved (ACC) |
|---|---|
| DeepSeek-R1 (single run) | 11/25 |
| DeepSeek-V3 ($14\times$ sampling) | 17/25 |

# G   Limitations

While this work advances *Neural Compilation* with a new benchmark and a self-evolving prompt optimization method, we also highlight several limitations.

**IR scope.**   Our study focuses on LLVM IR, for it is the most general Intermediate Representations, used by various programming languages, so the techniques are likely to generalize to other IRs, particularly those based on Static Single Assignment (SSA) form; however, generalization is not guaranteed. The design choices and assumptions in our method may not hold equally across all IR designs, especially those with significantly different semantics or control-flow structures.

**Benchmark scale.**   NeuComBack is moderate in size and not intended for pre-training or fine-tuning. It is designed primarily for evaluation: each task includes a verifiable test harness for automatic correctness (and, for L2, performance) measurement, and the dataset size keeps the cost/time of running frontier models practical and acceptable.

**Benchmark diversity.**   NeuComBack-L1 focuses on basic correctness; NeuComBack-L2 (TSVC-derived) emphasizes loop-centric, vectorization-friendly kernels that expose clear optimization headroom. This leaves less coverage of other compiler challenges (e.g., pointer-heavy code, irregular control flow, recursion, dynamic memory, interprocedural/code-layout optimizations, register pressure, and memory-hierarchy effects). Consequently, ACC+Perf gains on L2 should not be over-interpreted as universal across all optimization classes.

We will work to address these limitations in future work.

# NeurIPS Paper Checklist

1. **Claims**

   Question: Do the main claims made in the abstract and introduction accurately reflect the paper's contributions and scope?

   Answer: [Yes]

   Justification: The abstract and introduction accurately describe the paper's contributions, including the NeuComBack dataset, the baseline LLM evaluations, the proposed automatic prompt learning method, and the experimental results demonstrating its effectiveness in improving IR-to-assembly compilation. These are detailed in Sections 1 (Introduction), 3.2 (NeuComBack Dataset), 4 (Method).

   Guidelines:

   - The answer NA means that the abstract and introduction do not include the claims made in the paper.
   - The abstract and/or introduction should clearly state the claims made, including the contributions made in the paper and important assumptions and limitations. A No or NA answer to this question will not be perceived well by the reviewers.
   - The claims made should match theoretical and experimental results, and reflect how much the results can be expected to generalize to other settings.
   - It is fine to include aspirational goals as motivation as long as it is clear that these goals are not attained by the paper.

2. **Limitations**

   Question: Does the paper discuss the limitations of the work performed by the authors?

   Answer: [No]

   Justification: The paper does not have a dedicated "Limitations" section. However, potential limitations that could be discussed include:

   - **Scope of `NeuComBack` Benchmark:** The current `NeuComBack` benchmark primarily leveraging `ExeBench` and `TSVC` as data sources and currently focuses on the IR-to-assembly translation task. Future work could expand it with more data and additional compilation-related tasks (e.g., different IRs, other optimization stages).
   - **Extensibility of the Prompt Learning Method:** The proposed automatic prompt learning method has shown strong results. Nevertheless, it could be further extended by exploring more sophisticated prompt evolution strategies, potentially incorporating techniques like genetic algorithms or other search heuristics to navigate the prompt space more broadly.
   - **Computational Cost:** The automatic prompt learning process, involving multiple LLM inference calls for generation, self-debugging, and insight extraction, can be computationally intensive. The associated costs might pose a practical challenge for very large-scale prompt searches or when using highly expensive LLM APIs.
   - **Generalizability to Weaker Models:** The current method demonstrates significant effectiveness with recent, advanced LLMs (e.g., DeepSeek-R1). Its efficacy and the transferability of learned insights might be more limited when applied to smaller or less capable models, which would require further investigation.

   Guidelines:

   - The answer NA means that the paper has no limitation while the answer No means that the paper has limitations, but those are not discussed in the paper.
   - The authors are encouraged to create a separate "Limitations" section in their paper.
   - The paper should point out any strong assumptions and how robust the results are to violations of these assumptions (e.g., independence assumptions, noiseless settings, model well-specification, asymptotic approximations only holding locally). The authors should reflect on how these assumptions might be violated in practice and what the implications would be.
   - The authors should reflect on the scope of the claims made, e.g., if the approach was only tested on a few datasets or with a few runs. In general, empirical results often depend on implicit assumptions, which should be articulated.
   - The authors should reflect on the factors that influence the performance of the approach. For example, a facial recognition algorithm may perform poorly when image resolution is low or images are taken in low lighting. Or a speech-to-text system might not be used reliably to provide closed captions for online lectures because it fails to handle technical jargon.

- The authors should discuss the computational efficiency of the proposed algorithms and how they scale with dataset size.

- If applicable, the authors should discuss possible limitations of their approach to address problems of privacy and fairness.

- While the authors might fear that complete honesty about limitations might be used by reviewers as grounds for rejection, a worse outcome might be that reviewers discover limitations that aren't acknowledged in the paper. The authors should use their best judgment and recognize that individual actions in favor of transparency play an important role in developing norms that preserve the integrity of the community. Reviewers will be specifically instructed to not penalize honesty concerning limitations.

3. **Theory assumptions and proofs**

Question: For each theoretical result, does the paper provide the full set of assumptions and a complete (and correct) proof?

Answer: [NA]

Justification: The paper presents an empirical study with a new benchmark dataset (NeuComBack) and a novel methodology (automatic prompt learning) for assembly generation. It does not propose new theoretical results that would require formal assumptions and proofs. The problem formulation in Section 3.1 defines the task but does not introduce theorems.

Guidelines:

- The answer NA means that the paper does not include theoretical results.

- All the theorems, formulas, and proofs in the paper should be numbered and cross-referenced.

- All assumptions should be clearly stated or referenced in the statement of any theorems.

- The proofs can either appear in the main paper or the supplemental material, but if they appear in the supplemental material, the authors are encouraged to provide a short proof sketch to provide intuition.

- Inversely, any informal proof provided in the core of the paper should be complemented by formal proofs provided in appendix or supplemental material.

- Theorems and Lemmas that the proof relies upon should be properly referenced.

4. **Experimental result reproducibility**

Question: Does the paper fully disclose all the information needed to reproduce the main experimental results of the paper to the extent that it affects the main claims and/or conclusions of the paper (regardless of whether the code and data are provided or not)?

Answer: [Yes]

Justification: The paper describes the dataset creation process (Section 3.2), evaluation metrics (Section 3.2.3), LLMs used (Section 5.1), experimental setup including data splits (Section 5.1, 5.2), prompt learning parameters (epochs, batch size, self-debug rounds, Section 5.1), and architectures tested (x86_64, aarch64). Examples of baseline and learned prompts are also stated to be in Appendices A and B, which are crucial for understanding the inputs to the LLM.

Guidelines:

- The answer NA means that the paper does not include experiments.

- If the paper includes experiments, a No answer to this question will not be perceived well by the reviewers: Making the paper reproducible is important, regardless of whether the code and data are provided or not.

- If the contribution is a dataset and/or model, the authors should describe the steps taken to make their results reproducible or verifiable.

- Depending on the contribution, reproducibility can be accomplished in various ways. For example, if the contribution is a novel architecture, describing the architecture fully might suffice, or if the contribution is a specific model and empirical evaluation, it may be necessary to either make it possible for others to replicate the model with the same dataset, or provide access to the model. In general. releasing code and data is often one good way to accomplish this, but reproducibility can also be provided via detailed instructions for how to replicate the results, access to a hosted model (e.g., in the case of a large language model), releasing of a model checkpoint, or other means that are appropriate to the research performed.

- While NeurIPS does not require releasing code, the conference does require all submissions to provide some reasonable avenue for reproducibility, which may depend on the nature of the contribution. For example

(a) If the contribution is primarily a new algorithm, the paper should make it clear how to reproduce that algorithm.

(b) If the contribution is primarily a new model architecture, the paper should describe the architecture clearly and fully.

(c) If the contribution is a new model (e.g., a large language model), then there should either be a way to access this model for reproducing the results or a way to reproduce the model (e.g., with an open-source dataset or instructions for how to construct the dataset).

(d) We recognize that reproducibility may be tricky in some cases, in which case authors are welcome to describe the particular way they provide for reproducibility. In the case of closed-source models, it may be that access to the model is limited in some way (e.g., to registered users), but it should be possible for other researchers to have some path to reproducing or verifying the results.

5. **Open access to data and code**

Question: Does the paper provide open access to the data and code, with sufficient instructions to faithfully reproduce the main experimental results, as described in supplemental material?

Answer: [No]

Justification: We plan to release our dataset and code upon acceptance; currently no public repository is linked. We will include a link in the final version.

Guidelines:

- The answer NA means that paper does not include experiments requiring code.

- Please see the NeurIPS code and data submission guidelines (`https://nips.cc/public/guides/CodeSubmissionPolicy`) for more details.

- While we encourage the release of code and data, we understand that this might not be possible, so "No" is an acceptable answer. Papers cannot be rejected simply for not including code, unless this is central to the contribution (e.g., for a new open-source benchmark).

- The instructions should contain the exact command and environment needed to run to reproduce the results. See the NeurIPS code and data submission guidelines (`https://nips.cc/public/guides/CodeSubmissionPolicy`) for more details.

- The authors should provide instructions on data access and preparation, including how to access the raw data, preprocessed data, intermediate data, and generated data, etc.

- The authors should provide scripts to reproduce all experimental results for the new proposed method and baselines. If only a subset of experiments are reproducible, they should state which ones are omitted from the script and why.

- At submission time, to preserve anonymity, the authors should release anonymized versions (if applicable).

- Providing as much information as possible in supplemental material (appended to the paper) is recommended, but including URLs to data and code is permitted.

6. **Experimental setting/details**

Question: Does the paper specify all the training and test details (e.g., data splits, hyperparameters, how they were chosen, type of optimizer, etc.) necessary to understand the results?

Answer: [Yes]

Justification: Section 5 (Experiment) details the data splits for NeuComBack-L1 and NeuComBack-L2, the LLMs used (specifically DeepSeek-R1 for the core method evaluation), the number of epochs and batch sizes for prompt learning, the number of self-debugging rounds allowed per generation, and how the "learned prompt" was selected (best performing on the validation set). This provides a good understanding of the experimental procedure.

Guidelines:t

- The answer NA means that the paper does not include experiments.

- The experimental setting should be presented in the core of the paper to a level of detail that is necessary to appreciate the results and make sense of them.

- The full details can be provided either with the code, in appendix, or as supplemental material.

7. **Experiment statistical significance**

Question: Does the paper report error bars suitably and correctly defined or other appropriate information about the statistical significance of the experiments?

Answer: [No]

Justification: The paper reports performance metrics as percentages and raw counts (e.g., "28.00% (7/25)") in tables such as Table 1, 2, 3, 4, and 5. However, it does not include error bars, confidence intervals, or formal statistical significance tests for these results. Conducting multiple runs for each experimental condition to gather data for robust statistical significance analysis was deemed prohibitively expensive. This is primarily due to the high cost of LLM API calls, particularly when employing advanced reasoning models for complex, token-intensive tasks such as IR-to-assembly translation, where each sample can consume a significant number of tokens.

Guidelines:

- The answer NA means that the paper does not include experiments.
- The authors should answer "Yes" if the results are accompanied by error bars, confidence intervals, or statistical significance tests, at least for the experiments that support the main claims of the paper.
- The factors of variability that the error bars are capturing should be clearly stated (for example, train/test split, initialization, random drawing of some parameter, or overall run with given experimental conditions).
- The method for calculating the error bars should be explained (closed form formula, call to a library function, bootstrap, etc.)
- The assumptions made should be given (e.g., Normally distributed errors).
- It should be clear whether the error bar is the standard deviation or the standard error of the mean.
- It is OK to report 1-sigma error bars, but one should state it. The authors should preferably report a 2-sigma error bar than state that they have a 96% CI, if the hypothesis of Normality of errors is not verified.
- For asymmetric distributions, the authors should be careful not to show in tables or figures symmetric error bars that would yield results that are out of range (e.g. negative error rates).
- If error bars are reported in tables or plots, The authors should explain in the text how they were calculated and reference the corresponding figures or tables in the text.

8. **Experiments compute resources**

Question: For each experiment, does the paper provide sufficient information on the computer resources (type of compute workers, memory, time of execution) needed to reproduce the experiments?

Answer: [No]

Justification: The paper specifies the Large Language Models used (e.g., DeepSeek-R1, GPT-4o). It is clarified that model inference for these LLMs was performed via API calls. Consequently, the specific underlying compute hardware (e.g., GPU type, memory on the provider's side) for the inference step is managed by the API providers.

Guidelines:

- The answer NA means that the paper does not include experiments.
- The paper should indicate the type of compute workers CPU or GPU, internal cluster, or cloud provider, including relevant memory and storage.
- The paper should provide the amount of compute required for each of the individual experimental runs as well as estimate the total compute.
- The paper should disclose whether the full research project required more compute than the experiments reported in the paper (e.g., preliminary or failed experiments that didn't make it into the paper).

9. **Code of ethics**

Question: Does the research conducted in the paper conform, in every respect, with the NeurIPS Code of Ethics https://neurips.cc/public/EthicsGuidelines?

Answer: [Yes]

Justification: The research focuses on LLM-based code generation for compilers, using publicly available benchmarks (ExeBench, TSVC) to create a new dataset. It does not involve human subjects, sensitive personal data, or applications with immediate high-risk ethical concerns outlined in the NeurIPS Code of Ethics. The goal is to advance compiler technology.

Guidelines:

- The answer NA means that the authors have not reviewed the NeurIPS Code of Ethics.
- If the authors answer No, they should explain the special circumstances that require a deviation from the Code of Ethics.
- The authors should make sure to preserve anonymity (e.g., if there is a special consideration due to laws or regulations in their jurisdiction).

10. **Broader impacts**

Question: Does the paper discuss both potential positive societal impacts and negative societal impacts of the work performed?

Answer: [No]

Justification: The Introduction (Section 1) of the paper extensively discusses several potential positive societal impacts of Neural Compilation. These include its potential to streamline compiler development for novel architectures, reduce the time and effort needed for creating compilers for emerging Instruction Set Architectures (ISAs), facilitate the discovery of innovative optimization techniques, and transform how programming language researchers and hardware architects explore new designs. However, the paper does not explicitly discuss potential negative societal impacts. Examples of such impacts that could be relevant include the potential for deskilling in the compiler engineering workforce if traditional roles are significantly altered, the risk of misuse if the technology were adapted to generate obfuscated or malicious code (though this is not the paper's focus), or the environmental impact related to the energy consumption of training and deploying large LLMs for these compilation tasks.

Guidelines:

- The answer NA means that there is no societal impact of the work performed.
- If the authors answer NA or No, they should explain why their work has no societal impact or why the paper does not address societal impact.
- Examples of negative societal impacts include potential malicious or unintended uses (e.g., disinformation, generating fake profiles, surveillance), fairness considerations (e.g., deployment of technologies that could make decisions that unfairly impact specific groups), privacy considerations, and security considerations.
- The conference expects that many papers will be foundational research and not tied to particular applications, let alone deployments. However, if there is a direct path to any negative applications, the authors should point it out. For example, it is legitimate to point out that an improvement in the quality of generative models could be used to generate deepfakes for disinformation. On the other hand, it is not needed to point out that a generic algorithm for optimizing neural networks could enable people to train models that generate Deepfakes faster.
- The authors should consider possible harms that could arise when the technology is being used as intended and functioning correctly, harms that could arise when the technology is being used as intended but gives incorrect results, and harms following from (intentional or unintentional) misuse of the technology.
- If there are negative societal impacts, the authors could also discuss possible mitigation strategies (e.g., gated release of models, providing defenses in addition to attacks, mechanisms for monitoring misuse, mechanisms to monitor how a system learns from feedback over time, improving the efficiency and accessibility of ML).

11. **Safeguards**

Question: Does the paper describe safeguards that have been put in place for responsible release of data or models that have a high risk for misuse (e.g., pretrained language models, image generators, or scraped datasets)?

Answer: [NA]

Justification: The paper utilizes existing Large Language Models (e.g., DeepSeek-R1, GPT-4o) and introduces a benchmark dataset (NeuComBack) derived from public coding benchmarks. It does not release new foundational pretrained language models or image generators itself, nor does it use scraped datasets that would pose a high risk for misuse in terms of sensitive content or PII. The primary artifacts are the methodology and the benchmark.

Guidelines:

- The answer NA means that the paper poses no such risks.

- Released models that have a high risk for misuse or dual-use should be released with necessary safeguards to allow for controlled use of the model, for example by requiring that users adhere to usage guidelines or restrictions to access the model or implementing safety filters.
- Datasets that have been scraped from the Internet could pose safety risks. The authors should describe how they avoided releasing unsafe images.
- We recognize that providing effective safeguards is challenging, and many papers do not require this, but we encourage authors to take this into account and make a best faith effort.

12. **Licenses for existing assets**

Question: Are the creators or original owners of assets (e.g., code, data, models), used in the paper, properly credited and are the license and terms of use explicitly mentioned and properly respected?

Answer: [Yes]

Justification: The paper properly credits the creators of the datasets used: `ExeBench` (Armengol-Estapé et al., 2022) and `TSVC` (Maleki et al., 2011) are cited in Section 3.2.1. According to public sources, both datasets are released under open-source licenses, and their use in this paper complies with the respective licensing terms.

Guidelines:

- The answer NA means that the paper does not use existing assets.
- The authors should cite the original paper that produced the code package or dataset.
- The authors should state which version of the asset is used and, if possible, include a URL.
- The name of the license (e.g., CC-BY 4.0) should be included for each asset.
- For scraped data from a particular source (e.g., website), the copyright and terms of service of that source should be provided.
- If assets are released, the license, copyright information, and terms of use in the package should be provided. For popular datasets, `paperswithcode.com/datasets` has curated licenses for some datasets. Their licensing guide can help determine the license of a dataset.
- For existing datasets that are re-packaged, both the original license and the license of the derived asset (if it has changed) should be provided.
- If this information is not available online, the authors are encouraged to reach out to the asset's creators.

13. **New assets**

Question: Are new assets introduced in the paper well documented and is the documentation provided alongside the assets?

Answer: [Yes]

Justification: The paper introduces the NeuComBack dataset, and its construction, sources, levels, and metrics are documented in Section 3.2. The proposed automatic prompt learning method is described in Section 4. The paper itself serves as the primary documentation for these new assets. If these assets were to be released, further practical documentation (e.g., README files) would typically accompany them.

Guidelines:

- The answer NA means that the paper does not release new assets.
- Researchers should communicate the details of the dataset/code/model as part of their submissions via structured templates. This includes details about training, license, limitations, etc.
- The paper should discuss whether and how consent was obtained from people whose asset is used.
- At submission time, remember to anonymize your assets (if applicable). You can either create an anonymized URL or include an anonymized zip file.

14. **Crowdsourcing and research with human subjects**

Question: For crowdsourcing experiments and research with human subjects, does the paper include the full text of instructions given to participants and screenshots, if applicable, as well as details about compensation (if any)?

Answer: [NA]

Justification: The research described in the paper does not involve crowdsourcing experiments or research with human subjects.

Guidelines:

- The answer NA means that the paper does not involve crowdsourcing nor research with human subjects.
- Including this information in the supplemental material is fine, but if the main contribution of the paper involves human subjects, then as much detail as possible should be included in the main paper.
- According to the NeurIPS Code of Ethics, workers involved in data collection, curation, or other labor should be paid at least the minimum wage in the country of the data collector.

15. **Institutional review board (IRB) approvals or equivalent for research with human subjects**

Question: Does the paper describe potential risks incurred by study participants, whether such risks were disclosed to the subjects, and whether Institutional Review Board (IRB) approvals (or an equivalent approval/review based on the requirements of your country or institution) were obtained?

Answer: [NA]

Justification: The research does not involve human subjects and therefore did not require IRB approval or an equivalent review.

Guidelines:

- The answer NA means that the paper does not involve crowdsourcing nor research with human subjects.
- Depending on the country in which research is conducted, IRB approval (or equivalent) may be required for any human subjects research. If you obtained IRB approval, you should clearly state this in the paper.
- We recognize that the procedures for this may vary significantly between institutions and locations, and we expect authors to adhere to the NeurIPS Code of Ethics and the guidelines for their institution.
- For initial submissions, do not include any information that would break anonymity (if applicable), such as the institution conducting the review.

16. **Declaration of LLM usage**

Question: Does the paper describe the usage of LLMs if it is an important, original, or non-standard component of the core methods in this research? Note that if the LLM is used only for writing, editing, or formatting purposes and does not impact the core methodology, scientific rigorousness, or originality of the research, declaration is not required.

Answer: [Yes]

Justification: The use of Large Language Models (LLMs) such as DeepSeek-R1 is a fundamental and core component of this research. The entire paper is about "Neural Compilation," defined as using LLMs to generate assembly code, and proposes an "automatic prompt learning method" specifically to enhance LLM performance in this task. This is detailed throughout the paper, particularly in Sections 1 (Introduction), 3 (Neural Compilation Task), 4 (Method), and 5 (Experiment).

Guidelines:

- The answer NA means that the core method development in this research does not involve LLMs as any important, original, or non-standard components.
- Please refer to our LLM policy (`https://neurips.cc/Conferences/2025/LLM`) for what should or should not be described.

