# OpenReview forum: "QiMeng-NeuComBack: Self-Evolving Translation from IR to Assembly Code"
_NeurIPS.cc/2025/Conference — NeurIPS 2025 poster_

### Official Review · Reviewer_NLqo · 2025-06-30

**Clarity:** 3
**Significance:** 3
**Originality:** 3
**Rating:** 5
**Confidence:** 4

**Summary:**

The paper introduces a new benchmark dataset called NeuComBack for IR to Assembly neural compilation and evaluates it with state-of-the-art LLMs to establish baselines. It also proposes a self-evolving prompt optimization that lets LLMs improve by learning from self-debugging traces.

The method is validated in experiments with state-of-the-art models, where it is able to improve on the best performing baseline, DeepSeek-R1.

**Questions:**

- What is novel in the prompt learning?
- Is prompt learning also transferable on NeuComBack-L2, with respect to performance improvements?
- In Table 3, why was the accuracy not improved after two rounds?
- How was the correctness verification methodology of the TSVC dataset changed?
- How was the number of self-debug rounds determined? Why is it different in the different evaluations?
- Isn't the median of the middle 5 datapoints the same as just the median with an odd number of measurements?

**Ethical Concerns:**

["NO or VERY MINOR ethics concerns only"]

**Final Justification:**

I believe that the revisions will significantly strengthen the paper and address my concerns.

**Limitations:**

The paper doesn't discuss limitations. I think the chief limitations are the size of the dataset and the simple execution paths of the programs in it.

**Paper Formatting Concerns:**

No concerns.

**Quality:**

3

**Strengths And Weaknesses:**

## Strengths

The benchmarks use different state-of-the-art models, and the best performing model, DeepSeek-R1, is selected as the baseline. The presented method considerably improves upon it in both the correctness and performance of the compiled programs. Most of the correct programs (86.5%) surpassed clang-O3 performance.

The paper has ablation studies and interesting case studies of concrete optimizations where neural compilation surpasses the optimization of clang.

## Weaknesses

The Method section is very short and doesn't provide sufficient detail. It gives an idea about the prompt optimization, and the examples in the Appendix help fill some of the gaps, but one would be hard pressed to reproduce the method or evaluate its novelty based on this description. The whole workflow is mainly exposited in a Figure without detailed explanations.

The introduced dataset has two levels, but both levels have simple execution paths (the difference is in the loop intricacy because of data conditions). So as I understand there are no examples in the dataset that have more complex execution paths. The dataset seems to be specialized and not for general programming.

The introduced dataset is very small compared to ExeBench. It has hundreds of examples, ExeBench has millions. As a consequence, the test set is also very small, just 25 examples for NeuComBack-L2.

I don't understand why the Conclusion states that "This paper presents a novel compiler paradigm called Neural Compilation", because, as established in the Introduction and Related Work, Neural Compilation is an already existing and active field.

The word "performance" could be sometimes misunderstood, because performance evaluation could also refer to evaluating models. For example, on line 216-218: "Level 1 ... was not primarily designed for performance evaluation, making its programs unsuitable".

Typos:
- Figure 1 should not be positioned just after the section heading
- Typo in Figure 1: finial output
- line 269 is confusing, as O3 can be both an optimization level (meant here), but also an LLM.

---

> ### Author Rebuttal · Authors · 2025-07-31
>
> Thank you for your feedback. We have clarified our methodology and reproducibility, including the workflow, key hyperparameters, and open-source plans. We have given clearer descriptions of our dataset design, scale, and coverage. We corrected terminology and formatting for better clarity. We added new experiments on prompt transfer and cost-controlled evaluation. We also openly discussed limitations and plans to expand the dataset. These focused revisions directly address your main concerns and make our work clearer and stronger.
>
> ---
>
> ## W1: Methodological Clarity and Reproducibility
>
> Thanks for emphasizing the need for methodological detail and reproducibility. Our workflow is as follows:
>
> - For each mini-batch, an advanced LLM (e.g., DeepSeek-R1) generates assembly from IR using the current prompt, with up to N self-debug rounds.
> - Only successful self-correction traces are collected.
> - After each batch, a meta-prompt instructs the same LLM to extract error patterns and suggest prompt edits.
> - We always maintain a single evolving prompt—no genetic algorithm or population is used.
> - This runs for three epochs, and the prompt with the best validation accuracy is used for final testing.
> - All implementation details and meta-prompts are documented in the manuscript and appendix.
>
> We will provide comprehensive details and open-source our implementation to ensure transparency and reproducibility
>
> ---
>
> ## W2: Dataset Coverage and Generality
>
> Thank you for the opportunity to clarify the design intent of the NeuComBack benchmark. NeuComBack is structured with two complementary levels to jointly evaluate: (1) general-purpose functional correctness (L1, from ExeBench, covering a broad variety of real-world programs), and (2) advanced optimization performance (L2, from TSVC, focused on vectorization and loop optimization).
>
> L1 assesses breadth—an LLM’s ability to handle standard program logic across diverse constructs—while L2 targets depth, testing optimization strength on industry-standard kernels. We acknowledge that highly complex control flow (e.g., deep recursion, concurrency) is currently underrepresented, reflecting broader gaps in available datasets.
>
> Despite this, NeuComBack provides a strong foundational benchmark. As a next step, we plan to further expand the dataset to include more complex general-purpose programs, improving its coverage and value to the community.
>
> ---
>
> ## W3: Dataset Scale and Test Coverage
>
> Thank you for raising this important point about dataset scale. NeuComBack was designed for practical and rigorous evaluation—not pre-training—so it is moderate in size but high in quality and fully verifiable. We only include tasks from ExeBench with well-defined inputs/outputs and reliable test harnesses, ensuring correctness and performance can be automatically measured.
>
> We agree that scaling up is important. We are actively expanding NeuComBack with more sources, automated test harnesses, and greater diversity, which will further improve coverage and community value in future releases.
>
> We will clarify these points and roadmap in the revised manuscript. Thank you for your helpful feedback.
>
> ---
>
> ## W4: On the Use of “Novel Paradigm” in Conclusion
>
> Thank you for identifying this issue in our Conclusion. The reviewer is correct: Neural Compilation is an established field, and our work contributes to it rather than introducing it as a novel paradigm. The sentence "This paper presents a novel compiler paradigm called Neural Compilation" was an error and will be removed.
>
> To accurately reflect our contributions, we will revise the Conclusion to:
>
> *“This paper advances the field of Neural Compilation by introducing a dedicated IR-to-Assembly benchmark (NeuComBack) and a novel self-evolving prompt optimization method, enabling more systematic and rigorous evaluation of LLM-based compilation techniques.”*
>
> Our work builds on prior studies (see Section 2), providing a reproducible, extensible platform and new methods to support the Neural Compilation research community.
>
> We appreciate the reviewer’s careful reading and for helping us clarify and correctly position our contributions.
>
> ---
>
> ## W5: Clarification of “Performance” Terminology
>
> Thank you for highlighting the ambiguity in our use of "performance." We agree it is crucial to distinguish between a model’s task-solving accuracy and the efficiency of the generated code.
>
> To improve clarity, we will revise the manuscript to use:
>
> - “runtime performance,” “execution efficiency,” or “optimization quality” for code efficiency;
> - “functional correctness,” “success rate,” or “task accuracy” for model capability.
>
> These distinctions will be applied consistently, including in Section 3.2 and the Discussion. For example, lines 216–218 will be revised to:
>
> > “Level 1 … was not primarily designed to assess runtime performance or optimization quality, making its programs unsuitable for such evaluation.”
> >
>
> We appreciate the reviewer’s feedback and believe these changes will improve the precision and readability of our paper.
>
> ---
>
> ## Formatting, Typographical, and Clarity Issues (Typos questions)
>
> Thank you for carefully noting these formatting and typographical issues.
>
> - **Figure 1** will be moved to avoid following the section heading.
> - **Typo** “finial output” will be fixed to “final output.”
> - **"O3"** will be clarified throughout to indicate the compiler’s optimization level.
>
> We appreciate your attention to detail and will correct these points to improve clarity and professionalism.
>
> ---
>
> ## Q1: Novelty in Prompt Learning Approach
>
> Thank you for the question—here are the novel aspects of our prompt learning method:
>
> 1. **Selective Learning:** We use only self-debug traces from programs that were initially wrong but corrected, so the LLM learns effective correction strategies. Ablations confirm this is more data-efficient and achieves top results.
> 2. **Controlled Evolution:** Every prompt update is meta-verified by the LLM, ensuring changes are necessary and stable.
> 3. **Automatic Prompt Learning for Compilation:** We are among the first to show that LLMs can use their own self-debug history to improve correctness and optimization.
> 4. **Different from Prior Work:** Our method is fully automated and tailored for neural compilation, unlike earlier prompt optimization approaches.
>
> We will emphasize these contributions further in the revision.
>
> ---
>
> ## Q2: Transferability of Prompt Learning on NeuComBack-L2
>
> Thank you for asking about prompt transferability on NeuComBack-L2.
>
> we have curated a supplementary test set from our ongoing L2-extension efforts. This new set contains 24 programs (a scale close to original L2 test set) and was specifically chosen to cover a diverse range of categories beyond vectorization: control flow, floating-point operations, memory access patterns, and pointer arithmetic, derived from LLVM Test Suite.
>
> **Results (DeepSeek-R1):**
>
> - **Accuracy** improved from 32.0% to 41.7%.
> - **Performance**: Outperforming -O3 cases increased from 16% to 25% (after two optimization rounds).
> - **Debug rounds** dropped from 0.75 to 0.2.
>
> This shows our prompt learning generalizes well to new L2 tasks, improving both correctness and optimization. We will include this supplementary result in the revision.
>
> ---
>
> ## Q3: Interpretation of Stable Accuracy (ACC) Across Two-Phase Optimization in Table 3
>
> Thank you for raising this point about Table 3’s ACC metric.
>
> Our two-phase workflow is designed such that:
>
> - **Phase 1 (Generation + Self-Debug):** Produces functionally correct code. The “ACC” metric reflects the final success rate here.
> - **Phase 2 (Iterative Optimization):** Operates only on correct outputs from Phase 1, aiming to further improve runtime performance while maintaining correctness. ACC is not expected to change in this stage; instead, “ACC+Perf” indicates how many correct outputs also outperform the baseline.
>
> For context, with the **learned prompt**, 16 correct programs yielded 14 faster-than-O3 outputs (87.5%), while the baseline yielded 7/11 (63.6%).
>
> In summary, the stable ACC is expected, and the learned prompt boosts the chance of further optimizing correct programs. We will clarify these metrics and workflow in the revision.
>
> ---
>
> ## Q4: Changes to the Correctness Verification Methodology for TSVC
>
> Thank you for asking about our TSVC evaluation improvements.
>
> - **Input Initialization:** We now use random values (with a fixed seed) for array initialization, instead of fixed values. This helps reveal subtle errors and ensures reproducibility.
> - **Loop Iteration Verification:** We added an external loop counter (via the dummy function) to explicitly check that the required number of loop iterations is executed.
>
> These changes are included in our open-source evaluation framework for transparency and reproducibility.
>
> ---
>
> ## Q5: Determination of Self-Debug Rounds and Rationale for Differences
>
> Thanks for your question about self-debug rounds.
>
> - For **x86_64**, we fixed the max rounds at two for fairness and reproducibility.
> - For **aarch64**, we allowed up to four rounds, as it is a more challenging target for LLMs.
>
> This balances comparability (x86_64) and task difficulty (aarch64). We will clarify these settings in the revision.
>
> ---
>
> ## Q6: Clarification of “Middle 5” Median Performance Measurement
>
> Thank you for highlighting this subtle point about our performance measurement.
>
> Our process is based on execution order, not value:
>
> 1. **Chronological Filtering:** Each program is run 11 times; we discard the first 3 (warm-up) and last 3 (cool-down), keeping only the 4th–8th runs (“middle 5” by order).
> 2. **Stable Median:** We take the median of these 5 central runs to capture a robust, stable performance measurement.
>
> This approach helps minimize transient system effects and is standard in benchmarking. We will clarify this process in the revised manuscript. Thank you for prompting this clarification.

---

> ### Comment · Reviewer_NLqo · 2025-08-04
>
> Thank you for your detailed and insightful answers. I agree with almost all of them, except for the complexity of L1. In the paper you state that L1 "features C programs characterized by simple control flow structures", and "complex control flows, exemplified by nested for loops, are deliberately limited". I think this is a limitation which should be discussed.
>
> I believe that your modifications in reponse to all the reviewer's comments will significantly strengthen the paper, so I will raise my score.

---

> > ### Author Response · Authors · 2025-08-05
> > **Response to Reviewer NLqo’s Feedback**
> >
> > Thank you very much for your thoughtful feedback and for reconsidering our rebuttal!
> >
> > We sincerely appreciate your time and effort in evaluating our work. We are glad to hear that our responses have addressed your major concerns, and we truly value your positive recommendation. We also appreciate your suggestion regarding the limitation of the dataset, and will further discuss it in the revised version as you suggested.
> >
> > In addition, we would like to kindly remind you to update the review score accordingly. Please don't hesitate to let us know if you have any further concerns.

---

### Official Review · Reviewer_qGZW · 2025-07-01

**Clarity:** 4
**Significance:** 2
**Originality:** 2
**Rating:** 5
**Confidence:** 4

**Summary:**

The paper studies the task of neural compilation. Unlike past works in machine-learning for compilation, the compiler is not supported by a machine learning system that, for instance, predicts a good pass ordering for the compiler, but the compilation itself is carried out by a pre-trained language model. The specific task tackled is the stage of compiling LLVM IR to machine language.

Two benchmarks are sourced, one targeting compilation accuracy of IO-intensive real-world applications, the other for measuring both compilation accuracy and compiler optimization on compute-heavy functions from a compiler vectorization benchmark.

The paper establishes that:
- Reasoning models significantly outperform non-reasoning models on both benchmarks.
- A simple prompt evolution technique massively improves performance without any parameter updates.

**Questions:**

See above.

**Ethical Concerns:**

["NO or VERY MINOR ethics concerns only"]

**Final Justification:**

With a more detailed method section and additional experiments from the discussion period, the paper will improve further, so increasing my score by 1.

**Limitations:**

Limitations have not been addressed in the paper. (There is a section in the paper checklist on "what could have been discussed in the paper but wasn't".) It would improve the paper to state the obvious limitations clearly, e.g.:
- due to the small size of the dataset, it is difficult to use it for training,
- no studies on small models and SFT/RL,
- scope of the automatic prompt learning method is exactly for cases like this (reasoning-intensive, low-resource task).

**Quality:**

3

**Strengths And Weaknesses:**

**Strengths:**

1. The paper is clearly written and a pleasure to read. Congratulations!
2. The task of neural compilation is almost novel and considered at the right time when language models begin to be a viable option.
3. The proposed prompt learning technique is surprisingly effective, and calls for more intensive study of such techniques in reasoning-intensive, low-resource domains and potential integration into existing SFT and RL pipelines even.

**Weaknesses:**
1. The prompt learning technique lacks details. Please provide the exact update rule for the learned prompt (When/how often do you ask which LLM with which prompt to do what? Is there at any time exactly one prompt, or is it a simple genetic algorithm?) [The checklist says "Yes, reproducible", which isn't the case without such details in the appendix.]
2. The evaluation method lacks details. How is functional correctness evaluated? L206 mentions comparison of outputs, but on which inputs? Are the test cases model generated? Do they come with the originally sourced datasets? For TSVC, what is the "enhanced" correctness verification methodology (L208)? [Again, without, the paper is not reproducible as claimed in the checklist.]
3. The paper could benefit greatly from some additional analyses that shed more light onto what currently works and doesn't work with LLMs in this domain, for example:
- An inference scaling analysis that measures pass@k for different k for different model sizes, and similarly for the maximum number of self-debugging turns. Could there be a small cheap non-reasoning model that achieves similar scores as large reasoning models when comparing in a cost-controlled (compute-controlled) setup?
- Experiments on fine-tuning smaller open-weights models on sequences from large reasoning models.
- Generally, a comparison of SFT and RL to the automatic prompt learning technique.
- A comparison to other prompt learning techniques or genetic algorithms.
- Experiments on prompt transfer from a large reasoning model that figures out frequent mistakes and how to avoid them to smaller models / non-reasoning models that may be able to apply these rules.

[I do not request all these experiments, but stress that the paper could have benefitted from more results and analyses than the model comparison and the effectiveness of prompt learning.]

---

> ### Author Rebuttal · Authors · 2025-07-31
>
> Thank you for the detailed review and your constructive suggestions. To address your main concerns, we have provided precise details on our prompt learning and evaluation methods, conducted new experiments on cost-controlled comparison and prompt transfer as you suggested, and will add a dedicated Limitations section to the revised paper. We hope these additions strengthen the paper and address your questions. Please see our detailed responses below.
>
> ---
>
> # Weakness1: Details for Prompt Learning
>
> We thank the reviewer for highlighting the need for more precise details regarding our prompt learning process. Below we clarify when and how the prompt is updated, how updates are generated, why we use a single evolving prompt (not a genetic algorithm), and how we ensure stability in the update process.
>
> - Update rule and frequency: The prompt is updated after each mini-batch of compilation tasks during training. Specifically, after processing a batch, we collect self-debugging traces from those programs that were initially incorrect but corrected via self-debugging.
> - How the update is generated: We use the *same LLM* under evaluation to serve as a "prompt optimizer." The LLM is prompted (via a meta-prompt) to analyze these successful correction traces, extract error patterns and effective fixes, and propose edits to the existing prompt to reflect these new insights.
> - Single prompt, not genetic: At any time, there is a single evolving prompt; we do not maintain a population nor use a genetic algorithm.
> - Stability and confirmation: After the LLM suggests revisions, a comparison and confirmation step follows: we prompt the LLM to review the change list and confirm only the necessary and beneficial edits, adding stability to the evolution process.
>
> We will add these concrete implementation details to the revised Methods section for clarity. To further support reproducibility, we are preparing to publicly release both our benchmark and the full prompt learning framework code.
>
> ---
>
> # Weakness2: Details of Evaluation and Data Source
>
> We sincerely thank the reviewer for highlighting these important details regarding our evaluation methodology. We agree that precise descriptions are critical for reproducibility and appreciate the opportunity to clarify.
>
> ## 1. Inputs and Test Cases
>
> - All test cases are **sourced directly from the original ExeBench and TSVC benchmarks**; we do not use any model-generated test cases.
> - For each program, the assembly code produced by the LLM is assembled and executed using the standard input sets provided by the source benchmarks.
> - A program is considered functionally correct only if its output **exactly matches** the expected ground-truth output from the benchmark.
>
> ## 2. Enhanced TSVC Correctness Verification
>
> The TSVC benchmark mainly consists of array-based computation kernels evaluated through repeated loop operations. The original TSVC verification had two major limitations:
>
> - **Input Initialization:**
>     - Arrays were typically initialized with fixed values (e.g., all ones). This could cause certain computations (such as averaging) to yield the same result before and after the operation, potentially masking errors (e.g., the model fails to write back results but still passes correctness checks).
>     - **Our Enhancement:** We now initialize arrays with pseudo-random values generated using a fixed random seed. This provides a diverse set of initial conditions as test standards, while ensuring that results are fully reproducible.
> - **Loop Iteration Verification:**
>     - For timing accuracy, TSVC often runs functions for many iterations but does not always verify that the correct number of loop iterations are performed. In cases where single and multiple iterations yield the same result, an incorrect model implementation (with too few iterations) could go undetected.
>     - **Our Enhancement:** We use the original TSVC dummy external function as a hook and maintain an **external loop counter**. This allows us to check whether the generated code executes the intended number of iterations by comparing the observed count to the expected count.
>
> All of these improvements and the corresponding scripts will be included in our open-source evaluation framework for full transparency and reproducibility.
>
> ---
>
> # Weakness3: Cost-Controlled Model Comparison and Prompt Transfer
>
> You have provided several excellent suggestions for additional analyses that can shed more light on the capabilities and limitations of LLMs in this domain. We agree that these analyses significantly strengthen the paper. Following your suggestions, we have conducted two new experiments focusing on cost-controlled model comparison and prompt transfer, which correspond to your first and fifth points.
>
> ## 1. Cost-Controlled Model Comparison
>
> To address your question, we performed a comparison between DeepSeekR1 and DeepSeekV3.
>
> - Cost Analysis: We first estimated the relative inference cost and found that a single experimental run with DeepSeekR1 is approximately 14 times expensive than with DeepSeekV3.
> - Experiment Setup: Based on this finding, we conducted a cost-controlled evaluation. We allocated DeepSeekV3 a 14-fold computational budget, allowing it to perform more sampling, to match the cost of a single run of DeepSeekR1. All other experimental settings remain the same as those used in Section 5.2 on NeuComBack-L2.
> - Results: In our original experiments (Table 3), DeepSeekR1 solved 11 out of 25 problems (achieving functional correctness). However, under this new cost-controlled setting, DeepSeekV3 successfully solved 17 out of 25 problems.
>
>     | Setting | Problems Solved (Functional Correctness) |
>     | --- | --- |
>     | DeepSeekR1 | 11/25 |
>     | DeepSeekV3 (14x sampling) | 17/25 (+6↑) |
>
> This result directly validates the reviewer's suggestion. It demonstrates that under a fixed computational budget, a **non-reasoning model** can leverage its efficiency to afford more sampling attempts, ultimately outperforming a **reasoning-focused counterpart of the exact same parameter size**. Despite DeepSeekR1 and DeepSeekV3 sharing the same parameter count, this finding strongly suggests that the potential of a "small cheap non-reasoning model" to match "large reasoning models" in a cost-controlled setting is very achievable.
>
> ## 2. Prompt Transfer Across Models
>
> Inspired by your suggestion, we investigated whether the insights learned by a reasoning model could benefit a non-reasoning one.
>
> - Experiment Setup: We took the optimized prompt that was automatically learned by the large DeepSeekR1 model on the L2 training dataset, and applied this prompt to the DeepSeekV3 model without any other changes to the configuration as in Section 5.1.
> - Results: DeepSeekV3 with a basic prompt solved 4 out of 25 problems on the L2 test set. Remarkably, by simply providing it with the prompt learned by DeepSeekR1, DeepSeekV3's performance doubled, solving 8 out of 25 problems.
>
>     | Setting | Problems Solved (Functional Correctness) |
>     | --- | --- |
>     | DeepSeekV3 (basic prompt) | 4/25 |
>     | DeepSeekV3 (prompt learned by DeepSeekR1) | 8/25 (+4↑) |
>
> This finding provides strong evidence for the generalization performance of our proposed prompt learning method. The successful knowledge transfer demonstrates that the learned prompts are effective not only across different data distributions, but are also robustly transferable between different models. This capability significantly enhances the practical utility of our approach, as an effective prompt can be optimized once and then deployed across various models.
>
> ---
>
> # To Limitations
>
> Thanks for highlighting this important point. We fully agree that explicitly stating the key limitations of our work would provide a more balanced and honest perspective.
>
> In our revision, we will add a dedicated “Limitations” section that addresses the following points:
>
> - **Dataset Size:** Our benchmark is relatively small compared to large-scale code datasets, which makes it less suitable for fully supervised training of large models. Its primary role is for evaluation and targeted method development rather than model pretraining.
> - **Model and Method Scope:** Our experiments focus on large reasoning-capable models; we have not systematically studied small models or compared SFT/RL approaches, which would certainly provide further insights.
> - **Applicability:** The proposed automatic prompt learning method is designed for challenging, reasoning-intensive, low-resource tasks like IR-to-assembly, and its generalizability to other scenarios remains to be validated.

---

### Official Review · Reviewer_Yr1A · 2025-07-01

**Clarity:** 4
**Significance:** 2
**Originality:** 2
**Rating:** 2
**Confidence:** 4

**Summary:**

This paper tackles the challenge of using large language models (LLMs) to automate compiler development, a process known as Neural Compilation. They introduce NeuComBack a dedicated IR-to-assembly benchmark. Using that dataset they propose a self-evolving prompt optimization technique that lets the model iteratively refine its own prompts through self-debugging traces. In experiments on x86_64 and aarch64, this method raises functional correctness from 44% to 64% and from 36% to 58%, respectively, and yields assembly code that often outperforms clang-O3, demonstrating its promise for low-level neural compilation.

**Questions:**

1) Why aren't you using a bigger dataset from the community? AFAIK this should be "compile" and just lower that to assembly code?
2) How much time do you spend in optimizing a code using your approach how much time does it take to compile using O3?
3) Why didn't you use software that is being used to understand performance of systems on your L2 dataset, and you focused on the vectorization dataset?

**Ethical Concerns:**

["NO or VERY MINOR ethics concerns only"]

**Limitations:**

There is a limitation discussion in the appendix.

**Paper Formatting Concerns:**

Some margins with the tables are very narrow, making it hard to separate the caption from the main text.

**Quality:**

2

**Strengths And Weaknesses:**

The paper is well written and easy to follow.

Weaknesses:

The L1 dataset they propose is fairly limited. There exist way larger LLVM IR corpuses in literature that can be used to generate machine assembly by just invoking the LLVM pipelines (opt, llc). For example "Grossman, Aiden, et al. "Compile: A large ir dataset from production sources."".

The L2 dataset is interesting yet it is based on a pretty old publication (2011). The dataset is focused on vectorization, whereas compilers need to provide generic behavior. So using this as a datasets creates Neural Compilation that is effective for vectorization, an important optimization, but disregards other complex optimization opportunities. Nevertheless collecting performance data for the L2 dataset is a significant contribution.

I find the comparison with O3 misleading and some of the discussion around the purpose of compilation. Overall compilers need to be fast at compiling,  increase performance (execution time, binary size etc), be correct. A large part of the evaluation focuses on the last 2 parts and completely disregard the notion of "fast compilation". I think an interesting question would be provided a fixed compile time of "X" seconds which technology produces the fasters and correct binary. This is not addressed by this work and in my opinion it should be addressed by the "Neural Compilation" community overall.

---

> ### Author Rebuttal · Authors · 2025-07-31
>
> Thanks for your valuable feedback. We've addressed your comments on our dataset and compilation. We discussed and supplemented the experiment to address the concern. Please refer to the specific responses below and our rebuttal for more information.
>
> ---
>
> **Weakness 1 & Question 1.** Why not use existing dataset
>
> We thank the reviewer for this comment regarding our L1 dataset and for highlighting important large-scale IR datasets like "Compile". We agree that our benchmark's scale is a topic worth discussing, and we appreciate the opportunity to clarify our design philosophy.
>
> The primary goal of our benchmark is to provide a practical and effective tool for the **evaluation** of LLM-based compilation capabilities, rather than for large-scale pre-training or fine-tuning. This focus on evaluation has two main implications for the dataset's design:
>
> 1. **Practicality and Cost-Effectiveness:** For evaluation tasks, especially those involving powerful but costly models accessed via APIs (e.g., advanced reasoning models), a moderate, high-quality, and carefully curated benchmark is often more practical than a massive one. This approach keeps the financial and time costs of each evaluation cycle within a manageable range, which we believe encourages broader adoption and more rapid, iterative research in the community.
> 2. **Requirement for Verifiable Test Cases:** Another constraint is the requirement for verifiable test cases (i.e., defined inputs and corresponding expected outputs) to automatically assess the functional correctness of the generated assembly. While datasets like "Compile" offer a vast corpus of IR, **they do not provide the accompanying test cases needed for correctness verification**. Similarly, a significant portion of the original `ExeBench` dataset was unsuitable for automated testing. This necessity for complete, verifiable programming tasks, not just code snippets, greatly limited the pool of readily available data.
>
> At the same time, we agree with the reviewer that the current scale of our benchmark is a starting point, and your point is very constructive. In our future work, we plan to continuously improve the quality and expand the scale of `NeuComBack`. We will explore methods to incorporate more diverse programs and potentially semi-automatically generate test harnesses for larger code corpora. This will not only enhance its utility for evaluation but also pave the way for creating valuable resources for LLM training in the compilation domain.
>
> We will add a note in the revised manuscript to clarify this evaluation-centric design philosophy. Thank you for the valuable feedback.
>
> ---
>
> **Weakness 2 & Question 3.** Dataset Diversity
>
> We thank the reviewer for their valuable and constructive feedback on the L2 dataset, and for acknowledging the contribution of our performance data collection.
>
> The TSVC dataset we use mostly consists of multi-level loop-based array floating-point computations, which resemble the structure of complex hot functions in real systems, while remaining relatively simple. This dataset is used by many recent works, such as **LLM-Vectorizer**[1] and [2].  Moreover, it provides optimization opportunities in vectorization, revealing the potential for neural compilation. Therefore, despite appearing limited in diversity, our L2 dataset is a suitable starting point for neural compilation optimization research.
>
> At the same time, **we fully agree with your critical point**: a benchmark focused primarily on vectorization provides a limited view of a compiler's full capabilities. A comprehensive evaluation must extend to more general program structures and optimization types. We take this concern seriously and have, in fact, already begun work to address this. We plan to release an enhanced version of the dataset in the near future.
>
> This concern also raises a question about the generalizability of our findings: are the compilation strategies learned on the vectorization-heavy L2 dataset specific to that task, or do they have broader applicability in improving correctness and performance across diverse program structures?
>
> To address this, we have curated a supplementary test set from our ongoing **L2-extension efforts.** This new set contains 24 programs (a scale close to original L2 test set) and was specifically chosen to cover a diverse range of categories beyond vectorization: control flow, floating-point operations, memory access patterns, and pointer arithmetic, derived from [`LLVM Test Suite`](https://github.com/llvm/llvm-test-suite/tree/main/MultiSource). We then conducted a new experiment to test the generalization of our method. We took the exact same prompt that was learned exclusively on our original L2 training data and applied it to this new, diverse test set without any further prompt learning.
>
> The results, using the DeepSeek-R1 model, still demonstrate a strong positive transfer effect, over the baseline vanilla prompt:
>
> | Category | Metric | Improvement |
> | --- | --- | --- |
> | Functional Correctness | Success Rate | 32.0% → 41.7% |
> | Runtime Performance | Outperformed -O3 Cases | 16% → 25% |
> | Debugging Efficiency | Avg. Self-Debug Rounds Required | 0.75 → 0.2 |
> - Functional Correctness (ACC): On this new diverse set, the prompt learned from L2 data improved the success rate from a baseline of 32.0% to 41.7%.
> - Runtime Performance (ACC + Perf):  After two rounds of iterative optimization, the learned prompt guided the model to produce code that outperformed the -O3 baseline in 25% of cases, up from 16% without the prompt, demonstrating its effectiveness in optimizing diverse code.
> - Debugging Efficiency: For the successfully compiled programs, the average number of self-debug rounds required was significantly reduced from 0.75 to 0.2 (with a maximum of 2 rounds allowed).
>
> Note that if we directly use this dataset to self-improve the prompt, the improvement may be more effective.
>
> We hope the above explanation, supported by new experimental data, can alleviate your concerns regarding the L2 dataset's focus. We are grateful for this insightful question, as it has motivated us to more rigorously validate our findings and ultimately strengthen this work.
>
> [1]  LLM-Vectorizer: LLM-Based Verified Loop Vectorizer. CGO 2025.
>
> [2] All you need is superword-level parallelism: systematic control-flow vectorization with SLP. PLDI 2022.
>
> ---
>
> **Weakness 3 & Question 2.** Compilation Time
>
> We thank the reviewer for raising this crucial point about compilation time. This is a fundamental aspect of any compilation technology, and your questions rightly push us to clarify the positioning and purpose of our Neural Compilation approach.
>
> First, to directly answer Q2: a standard `clang -O3` compilation for programs in our benchmark is very fast, typically completing in **less than a second**. In contrast, current advanced LLM requires significantly more time in a single neural compilation process: approximately 0.5 **minute** for non-reasoning models and 5 **minutes** for reasoning models.
>
> While it's true that for most compilation tasks, compilation time is more important than small gains in runtime performance, there are scenarios where optimizing heavily-used core or hot functions, despite only marginal improvements, can be worthwhile. A typical example is CUDA kernels in LLMs, where even minor performance gains justify high optimization costs. Many compute libraries invest significant manual effort in writing these kernels, and super-optimization techniques like STOKE[1] or LENS[2] also demonstrate the willingness to spend substantial time searching for marginal improvements. These tools often require **hours** to find optimal instruction sequences for small code kernels. Similarly, searching for optimal compiler **phase ordering** using techniques like genetic algorithms (as seen in early research) can involve **hours or even days** of searching to yield performance gains, such as [3] and [4].
>
> Viewed in this context, our method is an automated approach that invests a moderate amount of time—far less than exhaustive search, but more than a standard compile—to discover novel, high-performance assembly code that general-purpose heuristics of `-O3` might miss. This investment is highly valuable in domains where runtime performance is paramount, such as in scientific computing, database query engines, or deep learning kernels. In these areas, a one-time optimization cost is easily amortized by the cumulative savings from repeated, faster executions.
>
> Regarding your suggestion of a fixed-time-budget comparison: this is an excellent and insightful framing. Given the current state of the technology, the outcome is quite clear:
>
> - For a **short budget** (e.g., < 10 seconds), `clang -O3` is the undisputed winner.
> - For a **longer budget** (e.g., > 10 minutes), Neural Compilation offers the *potential* to produce a superior binary in terms of runtime performance.
>
> This frames the choice not as a direct competition on all axes, but as selecting the right tool for the job: rapid iteration vs. achieving peak performance.
>
> In conclusion, we fully agree that our work does not address "fast compilation." We will revise our manuscript to more clearly position Neural Compilation's goal: not to be fast *at* compiling, but to surpass the optimization limit of current compilers (i.e.,  `clang -O3`). Thank you for pushing us to clarify this important distinction.
>
> [1] Stochastic superoptimization. ASPLOS 2013.
>
> [2] Scaling up Superoptimization. ASPLOS 2016.
>
> [3] Exhaustive optimization phase order space exploration. CGO 2006.
>
> [4] Exponentially Expanding the Phase-Ordering Search Space via Dormant Information. CC 2024.
>
> ---
>
> **To Paper Formatting Concerns**
>
> Thank you for pointing this out. We will revise the formatting in the final version to ensure that table layouts are clearer and more visually distinct from the main text, improving overall readability.

---

> > ### Author Response · Authors · 2025-08-06
> > **Follow-up Clarification from Authors**
> >
> > Dear Reviewer Yr1A,
> >
> > Thank you again for your valuable feedback and for taking the time to read our rebuttal. We have carefully addressed each of the concerns raised, adding new experiments, analyses, and clarifications in our responses. In particular, we:
> > 1. Provided the rationale for our evaluation-focused dataset design and added a new experiment on a more diverse test suite to demonstrate our method’s generalizability.
> > 2. Discussed in detail the positioning of Neural Compilation with respect to compilation speed and runtime performance.
> >
> > We would be very grateful for the opportunity to discuss if these additions have helped address your concerns. We welcome any further thoughts you might have.
> >
> > We sincerely appreciate your time and effort in reviewing our work.

---

### Official Review · Reviewer_7UhH · 2025-07-02

**Clarity:** 3
**Significance:** 3
**Originality:** 3
**Rating:** 5
**Confidence:** 4

**Summary:**

The authors highlight a gap in the literature regarding the generation of assembly by large language models, specifically a dedicated benchmark for the neural compilation paradigm. While there exists previous work that has targeted high-level to assembly (C-to-x86_64), in this work, the authors focus on LLVM Intermediate Representation to x86_64 and aarch64. To produce a benchmark, the authors consider two sources of problems: ExeBench and TSVC, with TSVC being more challenging and targeting performance evaluation rather than purely correctness by creating problems where vectorisation is expected. To assess how well LLMs perform, they focus on two metrics: functional correctness (modulo a test suite) and correctness with performance gain. The authors further argue that the ExeBench problems are insufficiently complex to measure performance gain; therefore, the second metric only applies to problems derived from TSVC. To improve/control the impact of the prompt, the authors also devise an Automatic Prompt Optimisation flow that uses debug information from previous runs to refine the LLM prompt for the task. The authors start from an initial prompt template, which is then gradually evolved by asking the LLM to abstract information from debug traces and incorporate it into the prompt. The authors evaluate five models: GPT-4o, O3-Mini, O1, DeepSeek-V3, and DeepSeek-R1. No model achieves a performance higher than 50%, suggesting the benchmark is difficult. Further, performance improvement is only observed for ~22% of the cases (33 out of 151). To demonstrate the value of the prompt evolution, the authors conisder the strongest performing model (DS-R1) and show that, on 40 samples that were kept as a test set, the model improves significantly from 50% functional correctness, to 80%; noteworthy here is that the model demonstrates the ability to perform novel vectorisation optimisations when using the evolved prompt in conjucture with two rounds of self-debug. This example, and an additional one where precomputing a constant allows merging multiple add instructions into a packed double-word add, were used as case studies to demonstrate the discovered optimisations.

**Questions:**

Q1. In the paper, you discuss the various cost functions that can be used for performance evaluation; however, it remains unclear which exact cost function was used for this work. Details in §3.2.3, line 220, suggest run-time recordings with appropriate burn-in (ramp-up/ramp-down). Is this the only component of the cost function, or were others, such as instruction numbers, some weighted instruction sum (for example, accounting for micro-code), etc., also used?

Q2. For functional correctness, my understanding from the paper is that it is done via differential testing against a known good compiler. How are the driving inputs chosen? Do you perform random testing? Fuzzying? Some other technique (for example, manually pre-selected edge case inputs)?

**Ethical Concerns:**

["NO or VERY MINOR ethics concerns only"]

**Final Justification:**

I believe the authors have reasonably addressed concerns raised and have reasonably defined and positioned their work. With the revisions discussed during the rebuttal phase, I believe this is a strong paper and maintain my positive score.

**Limitations:**

The authors properly address limitations from the machine learning side: does the learnt prompt learn transferable/general patterns? Is the self-debug mechanism useful for the task? And some software engineering limitations: does the result generalise to other architectures (or at least one other architecture)?, are the learnt optimisations interesting (the case study)?

One limitation, which exists by design, is not addressed. This work targets LLVM IR to assembly, and there may be a risk that the results do not generalise to other IRs. I believe this risk is minimal, and the results should generalise to other Static Single Assignment form IRs; however, this limitation should be mentioned in the work.

**Paper Formatting Concerns:**

Unless Figures should be strictly at the top of the page, or in-line wrap-formatted tables are discouraged, no concerns with regards to formatting.

**Quality:**

4

**Strengths And Weaknesses:**

+ S1:  Novel benchmark that is not saturated (best model is <50%)
+ S2: Targetting the LLVM IR allows the higher-level programs to come from a variety of languages as long as they provide an LLVM compiler (flang, clang, etc.)
+ S3: Example of a Domain Specific APO (though the claim in S2.2 is extremely specific (Line 128), making the novelty claim trivially true; I think there is merit in the approach, however, that paragraph made it _seem_ overly defensive.)
- W1: Some design decisions are unclear: why is the initial generation + iterative optimisation chosen? Is the output stable under iterations, i.e. are the changes by the LLM minimal in some sense (such as instructions changed)?
- W2: Missing details concerning input generation for functional correctness. I would guess equivalence testing under a fuzzing framework; however, I have to guess.

+/- (Mixed/S4/W3): They detail the experimental design for the runtime evaluation (performance) -- which is good methodology (+) -- but they do not provide the raw runtime results of the LLM snippets vs `clang -O3` in the Appendix.

---

> ### Author Rebuttal · Authors · 2025-07-31
>
> Thanks for your valuable feedback. We've addressed your comments on our workflow and evaluation. We discussed and supplemented the experiment to address the
> concern. Please refer to the specific responses below and our rebuttal for more information.
>
> ---
>
> **Weakness 1.** Global Workflow Design
>
> Our choice of a two-stage workflow involving "initial generation" followed by "iterative optimization" was driven by two primary considerations:
>
> - **Inspired by Compiler Design:** This approach draws an analogy to the design of traditional compilers, which separate code generation (lowering) from subsequent optimization passes. By decomposing the complex task of compilation, we can better guide the Large Language Model (LLM) to focus on distinct objectives.
> - **Reducing Task Complexity:** This separation is designed to reduce the cognitive load on the LLM of achieving high-quality compilation in a single step.
>     - In the **Initial Generation** phase, the model's primary goal is to ensure **functional correctness** by faithfully translating the IR into executable assembly code. This serves as the fundamental requirement.
>     - Building upon this foundation of correct code, the **Iterative Optimization** phase then focuses on enhancing **performance**. The model can make targeted improvements based on performance feedback.
>
> This separation of concerns makes the entire neural compilation task more tractable and effective for the LLM. It also allows for more precise analysis and improvement of the model's specific capabilities (i.e., whether it struggles with generation or optimization).
>
> ---
>
> **Weakness 1.** LLM Iterative Process
>
> In our current design, we do not explicitly constrain the scope of modifications the LLM can make (e.g., by limiting the number of instructions changed). However, our analysis of the optimization traces reveals that the LLM tends to perform incremental improvements and local adjustments on top of the existing correct code, rather than conducting radical rewrites.
>
> To quantify this observation, we calculated the average percentage of code lines modified by each model. The results are as follows:
>
> | Model | Average Change Ratio of Code Lines |
> | --- | --- |
> | GPT-4o | 23.09% |
> | O3-Mini | 26.39% |
> | O1 | 17.99% |
> | DeepSeek-V3 | 24.52% |
> | DeepSeek-R1 | 21.79% |
>
> This data indicates that the optimization process is **relatively stable**, even without explicit constraints. The average modification rate, ranging from approximately 17% to 27%, confirms that the process is an evolution of the existing code rather than a complete reconstruction.
>
> ---
>
> **Weakness 2 & Question 2.** Functional Correctness Evaluation
>
> We sincerely thank the reviewer for pointing out this missing detail regarding our evaluation methodology.
>
> For the portion of the dataset derived from ExeBench, we directly used the correctness test cases provided by ExeBench itself; we only modified the program format to make it suitable for our task. For the portion derived from TSVC, based on the original test cases in the dataset, we added loop iteration counting, and import random arrays to get more rigorous verification mechanism to ensure that the correctness reports are as accurate as possible, as is described below:
>
> Most of the test cases in the TSVC dataset involve loop-based computations on arrays. In the original validation setup, many examples initialize arrays with fixed values (e.g., all elements set to 1). This can lead to situations where, for certain operations such as averaging, the input and output arrays remain identical, making it impossible to detect specific types of model errors. To address this, we modified the initialization phase to assign array values using pseudorandom numbers generated with a fixed random seed. This ensures diverse initial values for testing computational correctness while maintaining reproducibility.
>
> On another front, TSVC improves timing accuracy by running the same function many times in a loop, but it does not record the actual number of iterations. As a result, for functions where the output is the same whether computed once or multiple times, the model might incorrectly generate a loop with insufficient iterations, and such errors would go undetected. To improve this, we utilize the external function calls used in TSVC as a hook to count iterations. We maintain an external counter that records how many times the loop is executed and use this information for validation.
>
> We will incorporate these specific details into the methodology section of our revised manuscript to ensure our verification process is clearly described.
>
> ---
>
> **Weakness 3.** Detailed Runtime Performance
>
> We thank the reviewer for suggesting that we include the detailed performance results. We agree that this data improves the paper's transparency and will add a corresponding table to the Appendix in the revised manuscript.
>
> The table below presents the speedup of our method's generated code relative to `clang -O3` for the 16 functionally correct programs in Table 3.
>
> The data shows that our method outperformed `clang -O3` in 14 of the 16 cases. The average speedup across these 16 programs is **1.28x**, with a maximum observed speedup of **3.25x**.
>
> | Program Name | Speedup (LLM over O3) |
> | --- | --- |
> | **s331** | **3.2485** |
> | **s1232** | **2.0621** |
> | **s332** | **1.9954** |
> | **s112** | **1.1048** |
> | **s152** | **1.0852** |
> | **s2251** | **1.0543** |
> | s451 | 1.0288 |
> | s319 | 1.0278 |
> | vdotr | 1.0277 |
> | s3112 | 1.0275 |
> | s4121 | 1.0274 |
> | s231 | 1.0253 |
> | s113 | 1.0247 |
> | s422 | 1.0224 |
> | s452 | 0.8621 |
> | s341 | 0.7924 |
>
> This detailed data will be included in the Appendix of our revised manuscript to provide a more complete performance overview.
>
> ---
>
> **Question 1.** Cost Function
>
> We thank the reviewer for this question seeking further clarification.
>
> To be precise, the **sole metric** used for the performance cost function, `c(·)`, in our work was the **program's execution time**. Our methodology for measuring this was to run each program 11 times and record the median **execution time of the middle 5 runs**. We adopted this approach to mitigate system noise and ensure that our performance results were stable and reliable.
>
> We did **not** incorporate other factors, such as instruction count, into our primary performance metric for this study. While these are all valid and interesting metrics, we chose execution runtime because it is the most direct and practical measure of the final performance experienced by a user, which was the primary focus of our evaluation.
>
> We will revise the manuscript to make this definition more explicit. Thank you for pointing out this need for clarification.
>
> ---
>
> **To Paper Formatting Concerns**
>
> Thank you for the concerns. We will make our best effort to improve the formatting in the final version to enhance the overall clarity and visual presentation of the paper.
>
> ---
>
> **To Limitations**
>
> We appreciate the thoughtful summary and highlighting an important limitation that we had not explicitly discussed.
>
> As noted, our study focuses on LLVM IR, for it is the most general Intermediate Representations, used by various programming languages, so the techniques are likely to generalize to other IRs, particularly those based on Static Single Assignment (SSA) form. We agree that this is a limitation worth stating clearly. The design choices and assumptions in our method may not hold equally across all IR designs, especially those with significantly different semantics or control-flow structures.
>
> We will revise the paper to explicitly acknowledge this limitation and clarify the scope of our current evaluation. We also appreciate the reviewer’s insight that the risk of non-generalization may be minimal in the SSA context, which helps contextualize the limitation.

---

> > ### Comment · Reviewer_7UhH · 2025-08-04
> >
> > I thank the authors for the detailed response.
> >
> > I think that this addresses all of my major concerns. Space permitting, S4.2.2 should contain perhaps a small sentence explaining the compiler passes inspiration for the online inference phase, as explained in W1 above in the response.
> >
> > The dataset issues highlighted by Yr1A will have to be considered; however, even under a very pessimistic reading, such as that the results only apply to programs that require vectorisation/SIMD optimisation, I remain positive about the paper.

---

> > > ### Author Response · Authors · 2025-08-05
> > > **Response to Reviewer 7UhH’s Feedback**
> > >
> > > Thank you very much for your positive feedback and for intending to recommend our manuscript for acceptance! We truly appreciate the time and effort you have dedicated to reviewing our work, as well as your valuable insights and constructive comments, which have greatly helped us improve the manuscript. Your support and recognition are sincerely appreciated.

---

### Note · Authors · 2025-08-13

Dear ACs and reviewers,

Thank you for your valuable feedback and careful oversight throughout the review process.

In our rebuttal, we have made earnest efforts to address the reviewers’ feedback through substantial new experiments and detailed clarifications. These efforts were positively received: Reviewer NLqo was convinced by our response and raised their score, and reviewer 7UhH also reaffirmed their positive stance. We believe we also comprehensively addressed the concerns from reviewer qGZW by running the new experiments they specifically suggested on cost-controlled comparison and prompt transfer.

Reviewer Yr1A (Score: 2) has not responded to our rebuttal. They raised three primary concerns: the dataset's scale, its specialization on vectorization, and our focus on runtime performance over compilation speed.
We clarified that our dataset is intentionally curated for rigorous, verifiable evaluation; we also ran a new experiment on a more diverse, general-purpose suite, which addressed the specialization concern and led reviewer NLqo to raise their score.
Reviewer 7UhH explicitly characterized the benchmark as "novel" and "not saturated"; they also noted the dataset scale concern raised by reviewer Yr1A, but considered it non-blocking.
On runtime vs. compile time, we explained from the perspective of neural compilation as a potential super-optimizer for performance-critical functions. These clarifications and results were well received by other reviewers, indicating the core issues have been addressed.

The reviewers agree that:
1. We have successfully addressed the reviewers' primary concerns regarding the evaluation's scope, generalization, and methodology through new experiments and detailed clarifications (Reviewers 7UhH, NLqo).
2. Our work introduces a dedicated, IR-to-ASM benchmark that is unsaturated and enables clean, comparable baselines across frontier models; reviewers highlighted its timeliness and value for the community (Reviewers 7UhH, qGZW, NLqo).
3. The proposed self-evolving prompt optimization method is considered novel and "surprisingly effective," providing a strong example of a domain-specific automatic prompt optimization technique (Reviewers qGZW, 7UhH, NLqo).

We look forward to incorporating these improvements into the final manuscript. Once again, thank you for your time and consideration.

Sincerely,

Authors of submission "Learning to Compile: Self-Evolving Translation from IR to Assembly Code"

---

### Decision · Program_Chairs · 2025-09-17

**Decision:**

Accept (poster)

**Comment:**

This work studies the extent to which a LLM can "compile" LLMV IR to machine code. It both evaluates a range of models on this task, showing that reasoning models are substantially better at it, and introduces a prompt evolution workflow that substantially increases compilation rates.

The reviewers generally agreed that this is a timely and novel contribution, the work is well-written, and the prompt evolution approach is remarkably effective. They noted some concerns that stemmed from unclear aspects in the writing. The authors addressed these thoroughly in the rebuttal.

Overall, the significant novelty and interesting results motivated the acceptance of this paper. Please ensure the final version includes all the suggested and discussed revisions.